# Adaptive force-position-speed collaborative process planning and roughness prediction for robotic polishing

**Ma Haohao[1,2], Azizan As'arry[2]\*, Niu Jing[1], Mohd Idris Shah Ismail[2],
Hafiz Rashidi Ramli[3], M. Y. M. Zuhri[4,5], Aidin Delgoshaei[6]**

1 School of Mechatronics and Automotive Engineering, Tianshui Normal University, Tianshui, China,
2 Department of Mechanical and Manufacturing Engineering, Faculty of Engineering, Universiti Putra
Malaysia, Serdang, Malaysia, 3 Department of Electrical and Electronic Engineering, Faculty of
Engineering, Universiti Putra Malaysia, Serdang, Malaysia, 4 Advanced Engineering Materials and
Composites Research Centre (AEMC), Department of Mechanical and Manufacturing Engineering,
University Putra Malaysia, Serdang, Malaysia, 5 Laboratory of Biocomposite Technology, Institute of
Tropical Forestry and Forest Product (INTROP), University Putra Malaysia, Serdang, Malaysia, 6 School
of Professional Studies, University of Kansas Edwards Campus, Overland Park, Kansas, United States of
America

\* zizan@upm.edu.my

IPN: Centro de Investigacion y de Estudios
Avanzados del Instituto Politecnico Nacional,
MEXICO

**Peer Review History:** PLOS recognizes the
benefits of transparency in the peer review
process; therefore, we enable the publication
of all of the content of peer review and
author responses alongside final, published
articles. The editorial history of this article is
available here: https://doi.org/10.1371/journal.
pone.0330979

## Abstract

In this study, an adaptive force-position-speed collaborative process planning framework for robot polishing was proposed to improve the stability of the robot polishing process. The material removal model based on Preston's theory was studied, and the factors of polishing pressure, tool speed, feed speed, and sandpaper type were considered to design the manual polishing experiment. The improved Dung Beetle Optimization algorithm, Back Propagation Neural Network, Finite Element Analysis, and Response Surface Methodology provide a strong guarantee for the selection of robot polishing process parameters. For curved workpieces, the curvature adaptive interpolation method is introduced to generate trajectories. An adaptive impedance control strategy is implemented to enhance force control, and PD iteration and RBF neural networks are used to ensure stable contact force and accuracy. The experimental results show that the root mean square error (RMSE) accuracy of the established roughness prediction model reaches 0.0001 μm, the proposed force control method is more stable, and the surface roughness is reduced by 20.79% on average compared to the baseline method, which proves the effectiveness of the framework in achieving high precision and high efficiency of robot polishing.

## 1. Introduction

As an integral part of automated manufacturing processes, robotic polishing plays a key role in improving the quality, efficiency, and precision of finished products. The manual polishing process is labor-intensive, time-consuming, and susceptible

**Data availability statement:** All relevant data are available from the Figshare repository at https://doi.org/10.6084/m9.figshare.29917520.

**Funding:** This work was supported in part by the Geran Putra Inisiatif (GPI) fund (GPI/2024/9794100). The funders had a role in the experimental study, data collection and analysis, and preparation of the manuscript.

**Competing interests:** The authors have declared that no competing interests exist.

to process variations. The introduction of robotic polishing addresses these challenges by providing a systematic and programmable approach to surface finishing. The application of robots to the workpiece grinding and polishing process ensures a higher level of uniformity in the final product and helps reduce production costs and increase overall productivity.

Many researchers have adopted robots for various industrial applications [1,2], and various polishing methods and technologies have been further developed. Z. Zhou et al. developed magnetic machining tools, studied the effect of adding mixed magnetic abrasives on grinding roughness, and used Ansys Maxwell software to simulate and analyze magnetic field-related parameters [3]. H. Wahballa et al. applied a 6DOF robot manipulator to perform robotic polishing on the violin surface, and studied how to achieve a smoother polishing trajectory and surface accuracy under different normal forces [4]. J. Zhang et al. studied the precise polishing process of industrial robot-assisted emery cloth wheels, taking into account parameters such as flexible contact and line spacing of the polished curved surface, and improving the surface roughness of the concave and convex surfaces [5]. Y. Wei et al. studied the structural design of the end effector of robot constant force polishing to ensure more precise tracking of the polishing contact force, and proved the rationality of the design through modeling and simulation [6].

The research mainly focused on the grinding process and grinding path, and continued to delve into the control of grinding and polishing contact force. G. Xiao et al. established an adaptive impedance compliance model for the robot polishing process, and studied the coupling relationship between force control and trajectory under tilt attitude [7]. D. Li et al. considered factors such as force, position, speed, and surface characteristics when studying robot polishing of curved workpieces, and applied compliant control to improve the polishing accuracy of curved workpieces [8]. J. Ge et al. used a laser vision sensor to obtain the surface characteristics of the workpiece to be polished, and proposed an adaptive speed grinding process, which can obtain better surface roughness than constant speed grinding [9].

At present, many scholars' research on automatic polishing technology mainly focuses on several aspects: First, the construction of a workpiece surface material removal model. The relevant parameters include polishing pressure, tool rotation speed, workpiece surface, feed speed, workpiece and tool materials, and Geometry, etc [10,11]. The second is the prediction study of the surface roughness of the workpiece to be polished. On the basis of establishing the prediction model, orthogonal experiments and other tests are carried out to establish the surface roughness model. This type of research has experimental guiding significance for automated polishing. The third is the research on polishing tool position trajectory planning, which mainly focuses on the surface characteristics of the workpiece to be polished and plans the polishing process parameters to further improve polishing accuracy and efficiency [12].

Results of reviewing the selected studies show that the application of robots in the grinding and polishing process allows for better automation and improved process results. With the continuous improvement of robot control technology, more stable,

convenient, and flexible robot polishing technology is becoming more and more widely used. The adaptability of robotic systems in response to different polishing requirements increases versatility, which is crucial in contemporary manufacturing environments. The ability to dynamically adjust force, position, and speed during polishing is critical to addressing the complexities of different materials, shapes, and surface properties. Despite the many advances in robotic polishing related technologies, many existing methods still face limitations in adapting to variable surface curvature, maintaining consistent polishing force under dynamic conditions, and optimizing process parameters to achieve surface quality. Traditional force control strategies often have problems with overshoot and poor anti-interference ability, while fixed trajectory planning methods have difficulty in dealing with the problem of uneven material removal on complex surfaces. Therefore, this study integrates curvature adaptive trajectory generation, a process planner based on improved dung beetle optimization, and an adaptive impedance control framework to overcome these challenges. The proposed method combines intelligent prediction, real-time adaptation and multi-factor optimization to provide a unified solution for high-precision robotic polishing.

Given these considerations, this study delves into the development of an adaptive force-position-velocity collaborative process planning framework for robotic polishing. By solving the nuanced challenges associated with robotic polishing, this research aims to contribute to the continued development of automated manufacturing processes, promoting quality control and efficiency improvements in different industrial sectors. This study has three contributions:

(1) To better adapt manual polishing processes to robotic polishing applications, the material removal mechanism in polishing was investigated. A mathematical model was developed, and optimal process parameters were obtained using the improved Dung Beetle Optimization (DBO) algorithm to guide manual polishing experiments in identifying the optimal parameters.

(2) To enhance the polishing accuracy of curved surfaces, a curvature-adaptive interpolation method was proposed for generating polishing trajectories. The effectiveness of the polishing process parameters and the stability of robotic polishing operations were validated using computer-aided engineering software.

(3) To enhance force control performance in robotic polishing, an adaptive impedance control method was proposed. The method's effectiveness was validated by measuring force control values during polishing and surface roughness values of the polished workpieces.

The content of this paper is as follows: Section 2 introduces the research methods of this paper in detail, including the mathematical model of the material removal mechanism of the polishing process, the polishing trajectory generation method with curvature adaptive interpolation, the adaptive impedance control method, the Design of Experiment (DOE), and the roughness prediction model of the Back Propagation Neural Network (BPNN). Section 3 introduces the experimental conditions and result analysis, including the experimental setup, surface roughness modeling, polishing process parameter research, and actual polishing experiments to verify the effectiveness and practicality of the method. Finally, Section 4 summarizes the main contributions of this study and puts forward suggestions for further research in the field of robotic polishing in the future.

## 2. Method

The research on robot polishing technology involves robot control, polishing tools, workpieces to be polished, polishing process parameters, and other aspects. This study mainly studies the material removal model, workpiece surface roughness, and process parameter optimization, surface adaptive speed optimization, and surface polishing trajectory planning in the process of curved workpiece polishing, in order to obtain better surface roughness results. The improvement method of the DBO algorithm involved in this paper will not be repeated here. The improvement method of the DBO algorithm used by the author has been publicly published in other papers.

The research workflow starts with the construction of curvature adaptive trajectory to ensure the accuracy of surface interpolation, especially in areas with large curvature. At the same time, a material removal model based on Preston's

theory is established to predict the relationship between process parameters and surface quality. Using this model, the optimal process parameters can be determined by heuristic algorithms. Subsequently, polishing experiments are carried out to verify the theoretical predictions, and the surface roughness prediction is performed using the improved BPNN model and experimental data. Finally, adaptive impedance control is used to improve the accuracy and stability of the robot polishing process. Fig 1 intuitively summarizes the entire workflow and illustrates the systematic approach adopted in this study.

## 2.1. Material removal model

When a surface normal polishing force is applied in the robotic polishing process, both the surface of the workpiece to be polished and the polishing tool will be deformed. In this study, only rigid polishing tool polishing is considered, and only

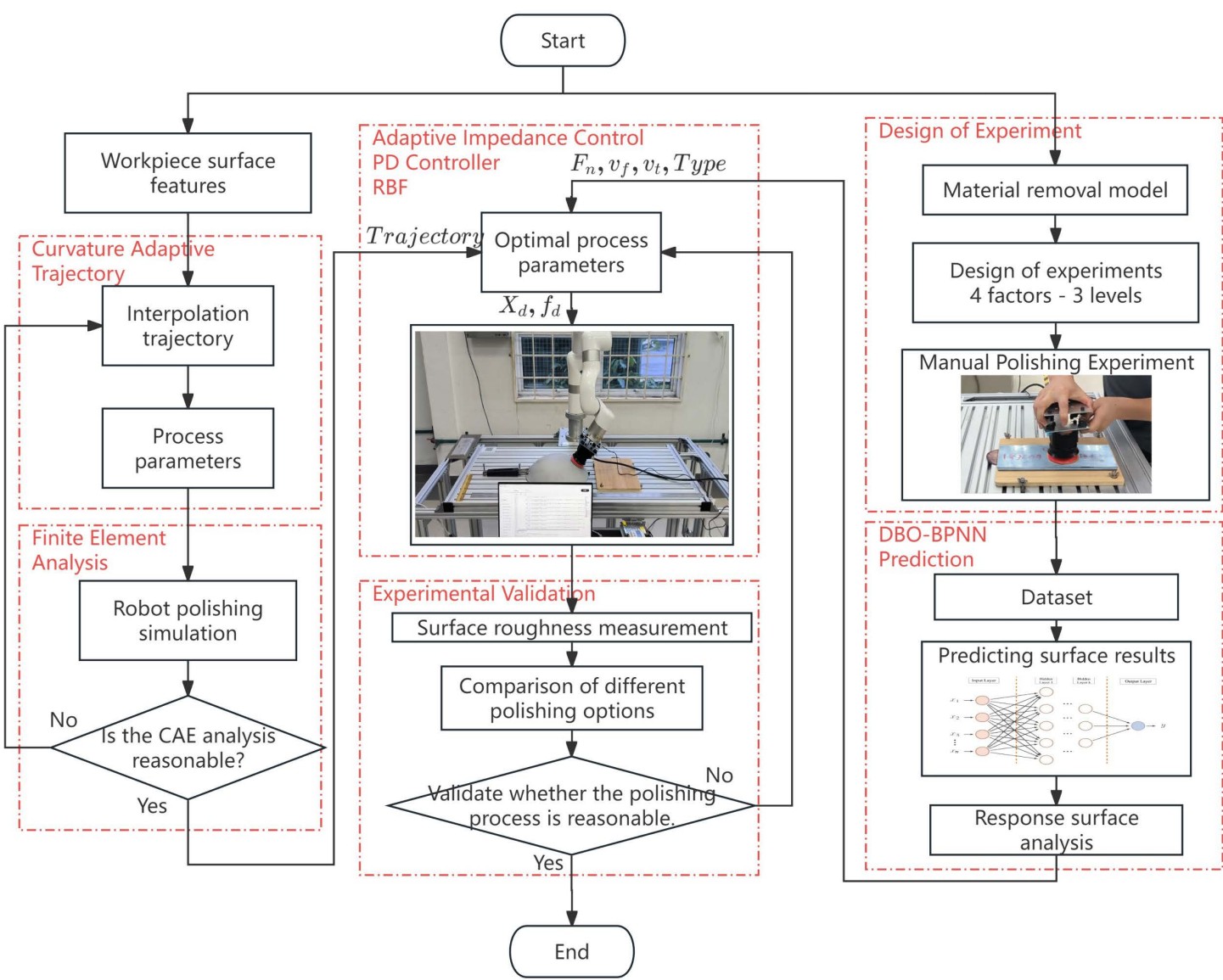

**Fig 1. Overall workflow of this study.**

the process parameters need to be optimized to obtain stable material removal. The process parameters that affect the contact situation during the polishing process include normal polishing force, feed speed, polishing tool rotation speed, workpiece surface characteristics, and material. Therefore, it is very necessary to study the material removal model, which can be used to obtain more uniform material removal and better polishing quality in the robotic polishing process [13].

Preston's theory [14] is used to establish a material removal model to analyze the robot polishing process. Preston's theory, such as formula (1), describes that when the polishing tool is in contact with the surface of the polished workpiece, the material removal depth $h(x,y,t)$ on the workpiece surface has a linear relationship with the pressure $p(x,y,t)$ of the contact point, contact time $dt$ and relative linear velocity $v(x,y,t)$.

$$h(x, y, t) = k \int_0^t p(x, y, t)v(x, y, t)dt$$

(1)

Where k is a proportional constant, which is related to factors such as the material of the polished workpiece and the polishing environment. This value can be fitted through experiments. The value of pressure $p(x,y,t)$ is equal to the polishing normal pressure $F(x,y,t)$ divided by the contact area $A(x,y)$ at that moment.

During the polishing process, it is calculated that only the polished workpiece produces elastic deformation. When the circular polishing tool contacts the polished workpiece surface, the surface contact area $A(x,y)$ can be regarded as an ellipse. As shown in the schematic diagram of Fig 2, z is the normal vector direction of the contact point on the workpiece surface, and a and b represent the values of the major and minor semi-axes of the elliptical contact. a is the X direction, b is the Y direction, and M is a certain micro-element area in the contact area [15].

The material removal depth of the contact micro-unit area M in Fig 2 is the material removed from the area from contacting the polishing tool to leaving the polishing tool (-b', b'), Update the Preston equation as formula (2). The relative linear velocity $v(x,y,t)$ can be determined by the tangential velocity $v_t(x,y,t)$ of the polishing tool rotation and the polishing tool feed speed $v_f(x,y,t)$, as formula (3).

$$h(x, y, t) = k \int_{-b'}^{b'} p(x, y, t)v(x, y, t)dy$$

(2)

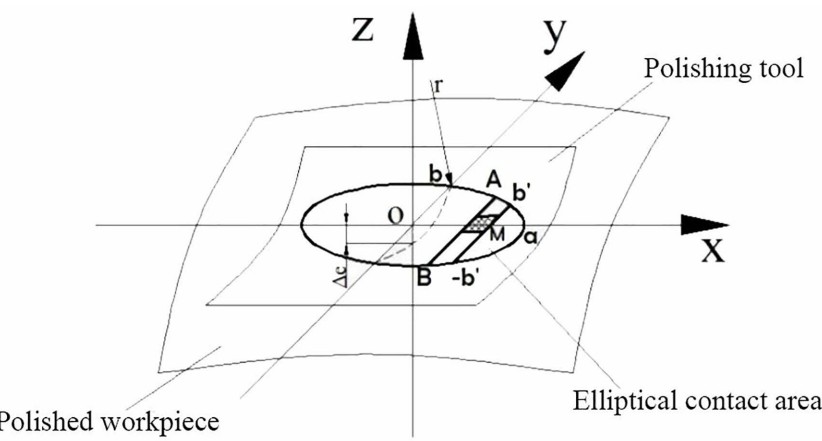

**Fig 2. Elliptical contact between polishing tool and workpiece surface.**

$$v(x, y, t) = \sqrt{v_t^2(x, y, t) + v_f^2(x, y, t)} \tag{3}$$

Equation (3) defines the relative velocity between the polishing tool and the workpiece. This formula is calculated using the Pythagorean theorem and assumes that the tangential velocity $v_t(x,y,t)$ of the rotating tool and the feed velocity $v_f(x,y,t)$ of the robot end effector are orthogonal components. In all experiments, the feed speed $v_f(x,y,t)$ is constrained to remain above a minimum threshold (5 mm/min) to avoid negligible motion and ensure meaningful material removal.

Equation (4) expresses the relationship between b' and the minor semi-axis b. Assuming that the pressure in the contact area in Fig 2 also obeys the elliptical distribution, $p_0$ is the pressure at the center point of the contact area, then the pressure at any point is calculated as in formula (5), where $Fn$ is the normal contact force at the center point during polishing.

$$b' = b\sqrt{1 - \left(\frac{x}{a}\right)^2} \tag{4}$$

$$p(x, y, t) = p_0\sqrt{1 - \left(\frac{x}{a}\right)^2 - \left(\frac{x}{b}\right)^2}$$
$$p_0 = \frac{3Fn}{2\pi ab} \tag{5}$$

Substituting Equations (3) and (5) into Preston Equation (2), formula (6) can be obtained.

$$h(x, y, t) = k\frac{3Fn\left(1 - \left(\frac{x}{a}\right)^2\right)}{4a}\frac{v_t(x, y, t)}{v_f(x, y, t)} \tag{6}$$

For a known surface $S$, it can be described as formula (7) by parameters u and v. The calculation of partial derivative is as formula (8). The surface normal vector $\boldsymbol{N}$ can be obtained by the cross product of two partial derivatives, such as formula (9).

$$S(u, v) = (x(u, v), y(u, v), z(u, v)) \tag{7}$$

$$\begin{cases} \boldsymbol{Su} = \frac{\partial S}{\partial u} = \left(\frac{\partial x}{\partial u}, \frac{\partial y}{\partial u}, \frac{\partial z}{\partial u}\right) \\ \boldsymbol{Sv} = \frac{\partial S}{\partial v} = \left(\frac{\partial x}{\partial v}, \frac{\partial y}{\partial v}, \frac{\partial z}{\partial v}\right) \end{cases} \tag{8}$$

$$\boldsymbol{N} = \boldsymbol{Su} \times \boldsymbol{Sv} \tag{9}$$

The partial derivatives of the normal vector $\boldsymbol{N}$ with respect to the parameters u and v are $\boldsymbol{Nu}$ and $\boldsymbol{Nv}$. Based on this, continue to calculate the curvature tensor $\boldsymbol{R}$ of the surface, as shown in formula (10). The main curvatures $r_1$ and $r_2$ of the surface are the eigenvalues of $\boldsymbol{R}$, which can be obtained by solving Equation (11), and $\boldsymbol{I}$ is the identity matrix.

$$\boldsymbol{R} = \begin{bmatrix} \frac{\|\boldsymbol{Nv}\|}{\|\boldsymbol{Su}\|} & \frac{\boldsymbol{Nu} \cdot \boldsymbol{Sv}}{\|\boldsymbol{Su}\| \cdot \|\boldsymbol{Sv}\|} \\ \frac{\boldsymbol{Nu} \cdot \boldsymbol{Sv}}{\|\boldsymbol{Su}\| \cdot \|\boldsymbol{Sv}\|} & \frac{\|\boldsymbol{Nu}\|}{\|\boldsymbol{Sv}\|} \end{bmatrix} \tag{10}$$

$$\det(\boldsymbol{R} - \lambda\boldsymbol{I}) = 0 \tag{11}$$

Equation (12) can calculate the relative Young's modulus $E^*$ during the polishing process, where $E_t$ and $E_w$ represent the Young's modulus of the polishing tool and the workpiece, respectively. $\lambda_t$ and $\lambda_w$ are the Poisson ratios of the polishing tool and the workpiece respectively.

$$\frac{1}{E^*} = \frac{1-\lambda_t^2}{E_t} + \frac{1-\lambda_w^2}{E_w}$$

(12)

The equivalent radius of curvature of the contact point during the polishing and grinding process of the calibration robot is $Re$, and its value can be obtained through experiments. It is obtained through the direction, principal curvature, and Young's modulus value of any contact point on the surface. Then the Preston Equation (6) can be sorted and simplified to obtain formula (13).

$$h(x,y,t) = k\frac{Fn^{\frac{2}{3}}}{Re^{\frac{1}{3}}} \frac{v_t(x,y,t)}{v_f(x,y,t)} \left(1 - \left(\frac{x}{a}\right)^2\right)$$

(13)

To more conveniently measure the parameters of the Preston equation of the material removal model in polishing, the position where the surface contacts $x=0$ is as shown in formula (14), which is the position of the maximum value of material removal.

$$h(0) = k\frac{Fn^{\frac{2}{3}}}{Re^{\frac{1}{3}}} \frac{v_t(x,y,t)}{v_f(x,y,t)}$$

(14)

To further discuss the correctness of the established Preston equation for material removal by polishing, and verify the relationship between material removal and speed, positive pressure, and material properties expressed in Equation (14), a finite element analysis was performed on curved surface polishing. Fig 3 shows the total deformation diagram (Fig 3A) and equivalent stress diagram (Fig 3B) obtained under this polishing condition. Rigid tools were selected in the polishing environment, the workpiece was made of aluminum alloy, and a normal contact force of 20 N was applied during the polishing process. Young's modulus is $7.1 \times 10^{10}$ Pa, Poisson's ratio is 0.33, and Bulk's modulus is $6.9608 \times 10^{10}$ Pa. The corresponding contact pressure was estimated based on the tool-workpiece contact area, which was approximated by the circular area of the polishing tool.

It can be seen that during the contact polishing process between the circular polishing tool and the irregular curved surface, the contact area is indeed approximately elliptical. The polishing process is similar to the diagram in Fig 2, and the entire process is completed from contact to departure along the direction of the long axis of the ellipse. It also verifies the correctness of Preston's theory and formula (14) applied in this study.

## 2.2. Curvature adaptive trajectory

The robot polishing process is realized through linear interpolation. The polishing trajectory can be regarded as the fitting curve of the tool contact point, which directly affects the polishing efficiency and the surface roughness of the workpiece. The traditional surface polishing trajectory interpolates the curve according to the equal residual constraint method. This study proposes a curvature-adaptive interpolation curve method, which can realize curve interpolation according to a given number of interpolation steps.

Currently, surface interpolation uses line-by-line interpolation of curves, and then completes the coverage of the entire surface. For curve interpolation, equal-spaced interpolation is often used [16-18]. The curve interpolation step length $L$ should satisfy the empirical formula (15). [$\varepsilon$] is the maximum chord height error allowed for the step length when polishing the curved surface. Exceeding this allowable error value will cause the roughness to become larger.

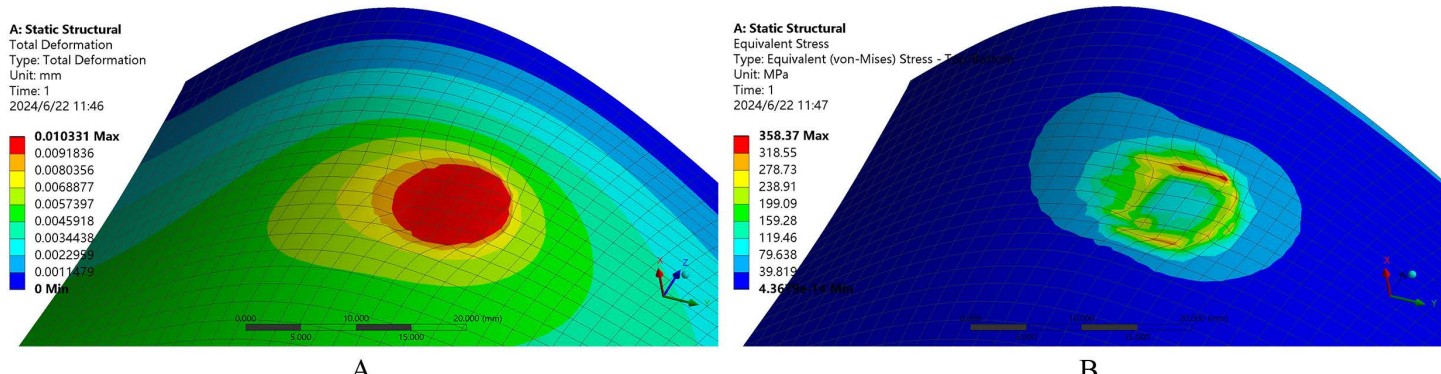

**Fig 3. Contact analysis of the curved surface polishing process (A) Total deformation, (B) Equivalent stress.**

$$L \leq \sqrt{8R[\varepsilon]} \tag{15}$$

When polishing with equal parameters, the initial step sizes of each polishing contact point are equal, such as the AB, BC, CD, DE, and EF segments in Fig 4. The chord height residuals corresponding to these step sizes are $\varepsilon_1$, $\varepsilon_2$, $\varepsilon_3$, $\varepsilon_4$, $\varepsilon_5$. In this case, the chord height error $\varepsilon_1$ corresponding to step BC and the chord height error $\varepsilon_4$ corresponding to DE exceed the allowable error $[\varepsilon]$. Add interpolation points between the two points BC and DE, and check the interpolation results of the curve point by point as shown in Fig 4.

This study proposes a new algorithm based on the traditional curvature step calculation. Fig 5 is the flow chart of this adaptive curvature interpolation calculation. Robot polishing targets two types of workpieces. One is a random workpiece. To achieve a better polishing effect, a 3D scanner can be used to obtain its surface point cloud model. The other is a workpiece with a known 3D digital model [19]. This surface data is more precise. After obtaining the surface data, extract the line-by-line curve interpolation point data. High-order spline curves can be used to interpolate curve values. In particular, point cloud models may have data point loss problems. Spline curve repair can make them smoother [20].

Solve the differential approximate derivative accumulation of the curve, and the maximum value is used as the maximum sequence number of the new point set to interpolate the new curve. The curvature of each point of the updated curve can be calculated. Special attention should be paid to the place with the largest curvature, and the error at this point is required to be less than the allowable error. If the error is greater than the allowable error, the number of interpolations for the new point set needs to be increased. Finally, a new curve interpolation sequence is obtained.

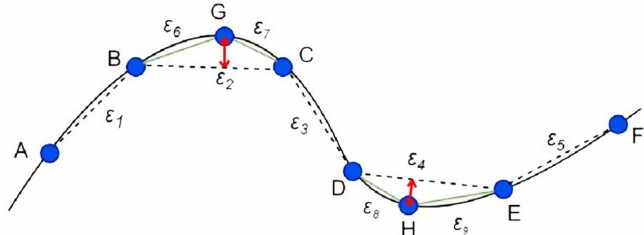

**Fig 4. Traditional curvature step optimization results.**

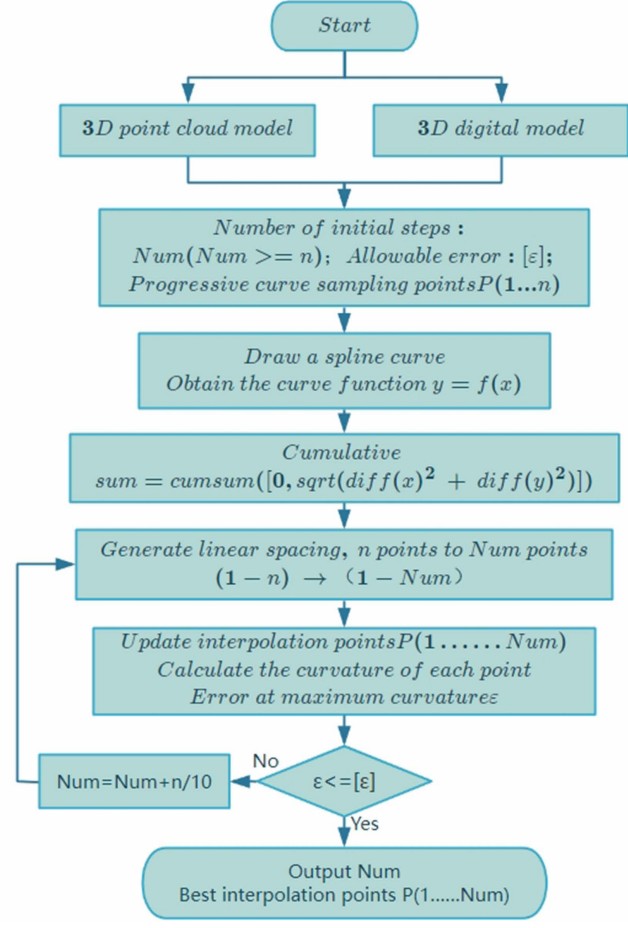

**Fig 5. Flowchart of curvature adaptive calculation.**

Given a curve consisting of 25 points, interpolation is performed according to the proposed curvature adaptive model. Fig 6 shows the superiority of this model. The spline curve in Fig 6 uses a quintic spline curve, and the corresponding function expression has been calculated.

Fig 6A-C are comparisons based on the same number of interpolation steps (25 steps), and Fig 6D is the interpolation result with an increased number of steps (35 steps). For a clearer comparison, we drew Fig 6C. The curvature values of each point are very small, so they were multiplied by 50. When the curvature is small and the curve is smooth, the step length is longer; when the curvature is large and the curve changes quickly, the step length is smaller. It can be seen that the proposed new interpolation model is more precise at the curvature than the original calculation model.

### 2.3. Adaptive impedance control

During the working process when the robot manipulator is in contact with the workpiece, the deformation error of the workpiece and the stiffness change of the robot manipulator itself always affect the force/position tracking accuracy of the robot manipulator. In order to enable the end effector of the robot manipulator to work in unknown environments, while tracking the preset trajectory and contact force, an impedance control model is adopted to ensure the accuracy and effectiveness of polishing and surface finishing [21]. The flowchart of the adaptive variable impedance control proposed in this study is shown in Fig 7.

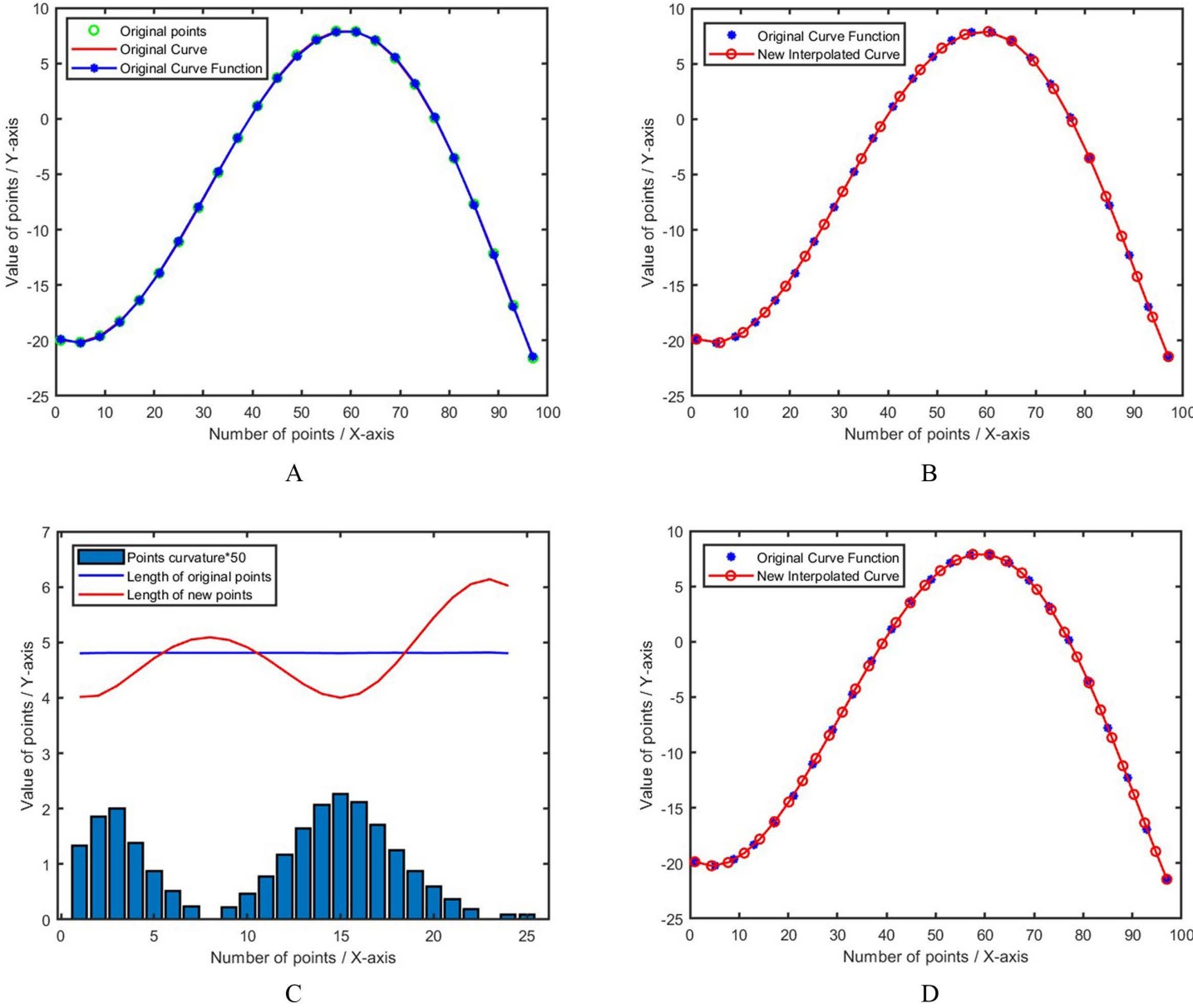

**Fig 6. Adaptive curvature interpolation results. A.** Set curve spline fitting, **B.** Adaptive curvature interpolation, **C.** Points step length and curvature, **D.** Increase the number of steps.

The PD iteration method is used to compensate for the stress and deformation bias of the workpiece, which improves the robustness of the impedance control method to the external environment. Compared with model-based control methods such as adaptive backstepping or sliding mode control, the iterative PD method achieves a good balance between computational simplicity and adaptability. It does not require an accurate dynamic model of the workpiece or contact surface and is particularly effective in practical robotic applications where the environment stiffness varies and there are uncertainties. In addition, its integration with the impedance control structure allows the system stiffness to be adjusted dynamically using the gradient descent method, thereby improving the transient force control performance without causing instability.

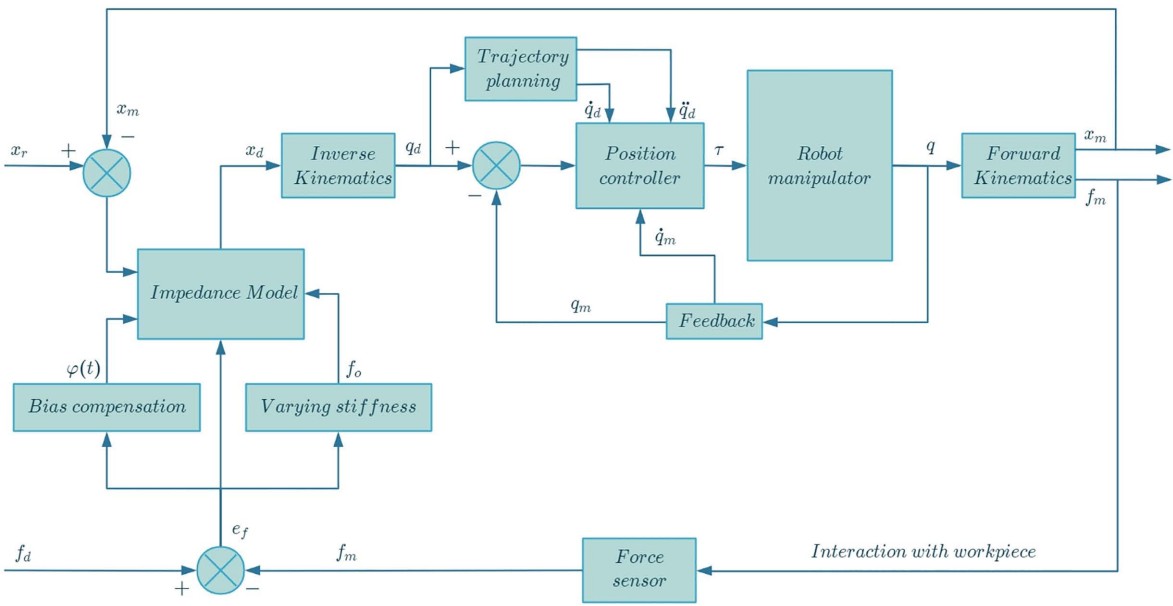

**Fig 7. Flowchart of adaptive variable impedance control.**

For the constantly changing stiffness of the robot manipulator, the contact force error is used as the optimization target, and then the impedance change is solved by the gradient descent method. The inverse kinematics is solved to obtain the joint parameters, and then the robot manipulator is controlled through the position control to complete the polishing [22].

**2.3.1. Impedance model.** The end effector of the robot manipulator contacts the workpiece to form a mass-spring-damper system. The impedance control model relates force and position to each other and realizes the relevant transformation of force and position, which can be expressed as Formula 16.

$$m\left(\ddot{x}-\ddot{x}_r\right)+b\left(\dot{x}-\dot{x}_r\right)+k\left(x-x_r\right)=f_d-f_m=e_f \tag{16}$$

Where, $m$, $b$, and $k$ denote the inertia coefficient, damping coefficient, and stiffness coefficient. $x$, $\dot{x}$, and $\ddot{x}$ denote the position, velocity, and acceleration of the actual trajectory of the end effector. $x_r$, $\dot{x}_r$, $\ddot{x}_r$ denote the position, velocity, and acceleration of the reference trajectory of the end effector. $x_m$, $\dot{x}_m$, $\ddot{x}_m$ denote the measured position, velocity, and acceleration of the end effector. $f_d$ is the desired contact force, $f_m$ is the value measured by the force sensor, and $e_f$ is the error value of the contact force.

Assuming that the robot manipulator end effector can accurately track the desired trajectory $x_d$, that is, $x = x_r$. The error between the measured value $x_m$ and the true value $\hat{x}$ is $\delta x$, that is, $x_m = \hat{x} + \delta x$. So, $x - \hat{x} = x_r - (x_m - \delta x) = e + \delta x$, and $e = x_r - x_m$. The stiffness change is $k_v$, and the impedance model can be expressed as formula (17). Then the bias compensation term is defined as $\phi(t)$, as shown in formula (18). After rearrangement, the impedance model that considers the measurement error and workpiece bias compensation can be obtained, as shown in formula (19).

$$m\left(\ddot{e}+\delta\ddot{x}\right)+b\left(\dot{e}+\delta\dot{x}\right)+(k+k_v)(e+\delta x)=f_d-f_m=e_f \tag{17}$$

$$\phi(t)=m\delta\ddot{x}+b\delta\dot{x}+k\delta x \tag{18}$$

$$m\ddot{e}+b\dot{e}+(k+k_v)e+\phi(t)=f_d-f_m=e_f \tag{19}$$

The iteration method is introduced to calculate the workpiece deviation compensation $\phi(t)$ in the external unknown environment during the robot working process, and the control law of $\phi(t)$ is designed as formula (20).

$$\phi(t) = \phi(t-\lambda) + \eta \left[ f_d(t) - f_m(t) \right] + \frac{\varsigma}{\lambda} \left\{ \left[ f_d(t) - f_m(t) \right] - \left[ f_d(t-\lambda) - f_m(t-\lambda) \right] \right\} \tag{20}$$

Where, the positive constant $\lambda$ is the sampling period of the system, and $\eta$ and $\varsigma$ are positive learning rates, $0 < \eta < 1$, $0 < \varsigma < 1$. Define the impedance change value $f_0 = k_v e$, then the impedance model can be written as formula (21). To further evaluate the dynamic behavior of the impedance control model, the position error term $e$ is differentiated with respect to time to derive the velocity and acceleration errors. Assuming that all relevant variables can be differentiated with respect to time, the first-order derivative of the position error is $\dot{e}$, and the second-order derivative is $\ddot{e} = \ddot{x}_d - \ddot{x}_e$. This allows the calculation of formulas (22) and (23).

$$m\ddot{e} + b\dot{e} + ke + \phi(t) + f_0 = f_d - f_m = e_f \tag{21}$$

$$\dot{e} = b^{-1} \left[ e_f - m\ddot{e} - ke - \phi(t) - f_0 \right] \tag{22}$$

$$\ddot{x}_d = \ddot{x}_e + m^{-1} \left[ e_f - b\dot{e} - ke - \phi(t) - f_0 \right] \tag{23}$$

An optimization control strategy is established with the control objectives of minimizing the contact force error, velocity tracking error, and trajectory position tracking error, and the optimization objective equation is selected as (24).

$$J(t) = \frac{1}{2} \left( \mu e_f^2 + v_1 \dot{e}^2 + v_2 e^2 \right) \tag{24}$$

The change in stiffness over time is tracked using the gradient descent method. The gradient of the optimization objective equation is calculated as shown in Equation (25). Here, $f_0$ represents the stiffness coefficient in the impedance model. It directly affects the system's compliance during contact. The gradient $\nabla J$ expresses the sensitivity of the force error for this stiffness, allowing adaptive adjustment to minimize $J(t)$. Here, it is assumed that the effect of $\dot{e}_f$ on $f_0$ is very small (the second-order effect can be approximately ignored). The third term in the gradient expression (25) is neglected due to its minimal sensitivity and weak coupling with the stiffness parameter.

$$\nabla J = \frac{dJ}{df_0} = \mu \cdot e_f \cdot \frac{\partial e_f}{\partial f_0} + \nu_1 \cdot \dot{e} \cdot \frac{\partial \dot{e}}{\partial f_0} + v_2 \cdot e \cdot \frac{de}{df_0}$$
$$\nabla J \approx -\mu \cdot e_f \cdot (x_d - x) \tag{25}$$

To minimize the contact force error and trajectory position error of the robot end effector, the change in stiffness $f_0$ changes in the opposite direction of the gradient as shown in formula (26), where $\alpha$ is the iterative calculation step size, $0 < \alpha < 1$.

$$f_0(t) = f_0(t-\lambda) - \alpha \nabla J \tag{26}$$

The impedance model is designed to follow the input reference trajectory and the desired force. An iterative algorithm is used to compensate for the external environment bias. The stiffness value in the impedance model is dynamically adjusted using the Newton gradient descent method, aiming to minimize the error of contact force and trajectory position, respectively.

**2.3.2. Trajectory position control.** The impedance model outputs the working trajectory $q_d$, which requires the robot position controller to implement trajectory tracking. In this study, a PI position controller was designed, and a Radial Basis Function (RBF) neural network was employed to approximate the unknown parts of the dynamic model. The RBF neural network, a type of feedforward neural network using radial basis functions as activation functions, is employed to approximate unknown dynamics in the control model. This approach enhances the trajectory tracking accuracy of the robotic manipulator, ensuring precise execution of the polishing process. The dynamic model of the robot manipulator can be expressed as formula (27).

$$\begin{cases} M_0(q)\ddot{q} + C_0(q, \dot{q})\,\dot{q} + G_0(q) + \tau_\Delta + f_n(\dot{q}) + \tau_d = \tau \\ \tau_\Delta = \Delta M(q)\ddot{q} + \Delta C(q, \dot{q})\,\dot{q} + \Delta G(q) \end{cases} \tag{27}$$

Where, $q$, $\dot{q}$, $\ddot{q}$ denote the angle, velocity, and acceleration of each joint of the robot manipulator. $M_0$, $C_0$, $G_0$ denote the positive definite inertia matrix, centrifugal force and Coriolis force matrix, and gravity matrix of the robot manipulator. $\Delta M$, $\Delta C$, $\Delta G$ denote the bias between the nominal value and the true value. $f_n(\dot{q})$ denotes the friction of the unknown joint. $\tau_d$ represents the unknown external disturbance. Define the state variables of the robot manipulator as $x_1 = q$, $x_2 = \dot{q}$, and the state space equation of position control is as shown in formula (28).

$$\begin{cases} \dot{x}_1 = x_2 \\ \dot{x}_2 = M_0^{-1}\left[\tau - \tau_\Delta - \tau_d - C_0 x_0 - G_0 - f_n\right] \end{cases} \tag{28}$$

Define the joint angle error of the robot manipulator as $e = q_d - x_1$, the joint angle velocity error is $\dot{e} = \dot{q}_d - x_2$. Define the generalized tracking error as $r = \dot{e} + \Lambda e$, where $\Lambda$ is a positive definite diagonal matrix. Taking the derivative of the generalized tracking error $r$ and multiplying both sides by $M_0$ get the Equation (29).

$$\begin{aligned} M_0\dot{r} = M_0\left(\ddot{e} + \Lambda\dot{e}\right) &= M_0\left(\ddot{q}_d + \Lambda\dot{e}\right) - M_0\ddot{q} \\ &= M_0\left(\ddot{q}_d + \Lambda\dot{e}\right) - C_0 r + C_0\left(\dot{q}_d + \Lambda e\right) + G_0 + \tau_\Delta + \tau_d + f_n - \tau \end{aligned} \tag{29}$$

Design the PI position control law as formula (30), where $K_P$ and $K_I$ are both positive definite diagonal matrices.

$$\begin{aligned} \tau =\ & M_0\left(\ddot{q}_d + \Lambda\dot{e}\right) + C_0\left(\dot{q}_d + \Lambda e\right) + G_0 \\ & + \tau_\Delta + \tau_d + f_n(\dot{q}) + K_P r + K_I \int r\, dt \end{aligned} \tag{30}$$

PI position control law includes error terms $\tau_\Delta$, external disturbance terms $\tau_d$, and friction terms $f_n(\dot{q})$. The LuGre model provides a detailed representation of friction dynamics by capturing both static and dynamic friction behaviors. The model describes friction as a combination of Coulomb friction, viscous friction, and a dynamic state variable representing the microscopic bristle deformation between contact surfaces. The LuGre friction model can be expressed as formula (31).

$$\begin{cases} f_n(\dot{q}) = \sigma_0 z + \sigma_1 \dot{z} + \sigma_2 \dot{q} \\ \dot{z} = \dot{q} - \sigma_0 \dfrac{|\dot{q}|}{g(\dot{q})} z \\ g(\dot{q}) = F_c + (F_s - F_c) e^{-(\dot{q}/v_s)^2} \end{cases} \tag{31}$$

Where, $\sigma_0$, $\sigma_1$, and $\sigma_2$ are model parameters, $z$ is the internal state variable, $Fs$ denotes the static friction torque, $Fc$ denotes the Coulomb friction torque, and $v_s$ denotes the Stribeck velocity. By utilizing the LuGre model, the controller can

accurately estimate and compensate for the friction forces. To improve the control performance, an RBF neural network was employed to approximate the unknown components of the control law. The newly designed RBF-PI position controller can be expressed as formula (32).

$$\tau = M_0(\ddot{q}_d + \Lambda \dot{e}) + C_0(\dot{q}_d + \Lambda e) + G_0 + f_n(\dot{q})$$
$$+ K_P r + K_I \int r dt + \hat{W}^T h(\theta) + K_r sgn(r)$$

(32)

The number of hidden layer nodes of the RBF neural network is selected as $N$. $\hat{W} \in \mathbb{R}^{N \times n}$ is the weight value of the neural network, which is used to approximate the optimal weight $W^* \in \mathbb{R}^{N \times n}$, and the weight error $\widetilde{W} = \hat{W} - W^*$. $h(\theta)$ is the output of the RBF function, $\theta = [e, \dot{e}, q_d, \dot{q}_d, \ddot{q}_d]^T$ is the input of the neural network, and $\varepsilon \in \mathbb{R}^n$ denotes the approximation error with an upper bound. The positive definite diagonal matrix $K_r > \|\varepsilon\|$. The adaptive law of the designed RBF neural network is $\dot{\hat{w}} = \Gamma^{-1} h r^T$. Where $\Gamma$ is a positive definite diagonal matrix.

### 2.3.3. Stability proof.
To further prove the stability of the designed control rate, evaluate the error energy of the system and the estimation error of the neural network weight, the Lyapunov function of formula (33) is selected [23].

$$V = \frac{1}{2} e^T M_0 e + \frac{1}{2} \widetilde{W}^T \Gamma^{-1} \widetilde{W}$$

(33)

According to the generalized travel error $r = \dot{e} + \Lambda e$, combined with $M_0$, $M_0 \dot{e}$ can be obtained, as shown in formula (34). Substituting the control law (32) into (34) and rearranging the terms, we can obtain formula (35), where $\in$ represents the approximation error of the RBF neural network.

$$M_0 \dot{e} = \tau - C_0 \dot{q} - G_0 - f_n(\dot{q})$$

(34)

$$M_0 \dot{e} = -K_P r - K_I \int r dt - K_r sgn(r) + \hat{W}^T h(\theta) - \in$$

(35)

Taking the time derivative of the Lyapunov function $V$ and substituting the above formula into it, we can obtain $\dot{V}$, as shown in formula (36).

$$\dot{V} = e^T M_0 \dot{e} + \widetilde{W}^T \Gamma^{-1} \dot{\widetilde{W}}$$

(36)

$$\dot{V} = e^T \left( -K_P r - K_I \int r dt - K_r sgn(r) + \widetilde{W}^T h(\theta) - \in \right) + \widetilde{W}^T \Gamma^{-1} \dot{\widetilde{W}}$$

(37)

Using the weight adaptation law $\dot{\widetilde{W}} = \Gamma^{-1} h(\theta) r^T$ of the neural network, $\widetilde{W}^T \Gamma^{-1} \dot{\widetilde{W}} = -\widetilde{W}^T \Gamma^{-1} h(\theta) r^T$ can be obtained. Therefore, the derivative of the Lyapunov function can be simplified to formula (38).

$$\dot{V} = -r^T K_p r - K_r \|r\|_1 + \in$$

(38)

Where, $\in$ denotes the approximation error of the RBF neural network, $\|r\|_1$ represents the L1 norm of vector $r$, also known as the absolute value norm. In robot control, by properly choosing the values of $Kp > 0$ and $Kr > 0$, we can ensure that the approximation error $\in$ of the RBF neural network is bounded. According to the Universal Approximation Theorem, there exists a constant $C > 0$ such that $|\in| < C$. Thus, we obtain the formula (39).

$$\dot{V} = -\|r\|^2 - K_r \|r\|_1 + C \tag{39}$$

Where, $C$ is a small constant representing the approximation error of the RBF neural network. By choosing sufficiently large control gains $Kp>0$ and $Kr>0$, we ensure that the negative terms dominate $C$, i.e., $\|r\|^2 + K_r \|r\|_1 > C$.

Therefore, $\dot{V} < 0$, and the Lyapunov function $V(t)$ is strictly decreasing. According to LaSalle's Invariance Principle, all trajectories converge to the largest invariant set where $\dot{V} = 0$, which implies $r \to 0$ as $t \to \infty$. Thus, the system is asymptotically stable.

**2.3.4. Parameter setting of the controller.** The impedance control parameters were selected based on previous robot polishing research and tuning experimental experience: m = 0.15 kg, b = 30 Ns/m and k = 500 N/m. The iterative update law for environmental deviation compensation adopted a system sampling period λ = 0.01 s, a forward learning rate η = 0.05 and ς = 0.05, and a gradient reverse iteration calculation step size α = 0.001. While the xArm platform introduces minor timing variations due to communication and internal processing delays, the effective update period in our experiments was observed to remain close to λ = 0.01 s.

The proportional gain and integral gain used in the PI position controller were $K_P = $ **diag**(30, 30, 30, 20, 20, 2020,30) and $K_I = $ **diag**(5, 5, 5, 3, 3, 33,5), respectively, which were determined by iterative adjustment to balance steady-state tracking and responsiveness.

The RBF neural network consists of 3 input nodes (joint position errors), 1 hidden layer with 10 radial basis neurons using a Gaussian activation function, and 1 output node. The radial function expansion is set to 1.5. The learning rate for weight adaptation is set to 0.01, and the initial weights are randomly initialized from a uniform distribution in the range [−0.5, 0.5].

To estimate the friction terms $f_n(\dot{q})$, the parameters of the LuGre friction model are selected as shown in Table 1.

## 2.4. Design of experiments

Robots are widely used to replace workers in polishing workpieces, especially in industrial production. The surface roughness of the workpiece is affected by multiple factors, mainly the workpiece material, polishing tool material, process parameters, and environmental equipment factors. Different materials have different densities, hardness, and cutting properties, which affect the method of polishing to remove materials. The polishing tool generally refers to the size parameters of the abrasive particles. The controllable factors in the experiment are mainly polishing pressure, polishing speed, feed speed, and polishing trajectory.

To systematically study the influence of factors on the roughness of the workpiece polished by workers using robots, the parameters of key factors were selected as experimental variables. A total of four factors include polishing positive force $Fn$ (N), tool speed $v_t$ (r/min), feed speed $v_f$ (mm/min), and sandpaper type $Tp$ (No.). Three levels were set for each variable in the experiment, as shown in Table 2.

**Table 1. Four factors and three levels of the experiment.**

| Parameter | Value | Unit | Description |
|---|---|---|---|
| $\sigma_0$ | 84000 | Nm/rad | Stiffness coefficient |
| $Fs$ | 8.16 | Nm | Static friction torque |
| $Fc$ | 3.82 | Nm | Coulomb friction torque |
| $\sigma_1$ | 260 | Nm·s/rad | Damping coefficient |
| $\sigma_2$ | 28 | Nm·s/rad | Viscous friction coefficients |
| $v_s$ | 0.0125 | rad/s | Velocity |

**Table 2. Four factors and three levels of the experiment.**

| Levels | Factors | | | |
|---|---|---|---|---|
| | *Force Fn (N)* | *Tool speed $v_t$ (r/min)* | *Feed speed $v_f$ (mm/min)* | *Sandpaper type Tp (No.)* |
| 1 | 15 | 5000 | 120 | 240 |
| 2 | 22.5 | 7500 | 160 | 400 |
| 3 | 30 | 10000 | 200 | 800 |

For the 4-factor 3-level experiment designed in this study, a comprehensive factorial experimental design would require 81 sets of experiments. In order to conduct the experiment more efficiently, $L_{27}$ (3 4) sets of simplified experiments were selected.

The orthogonal experiment designed in this way can ensure that each level of each factor is evenly distributed in the experiment, and can reflect the level changes of each factor as comprehensively as possible. After conducting 27 groups of experiments, range analysis and variance analysis can be performed to determine the main factors affecting the surface roughness of robotic polished workpieces, providing a data set for subsequent roughness prediction.

## 2.5. DBO-BPNN roughness prediction model

Back Propagation Neural Network (BPNN) is a relatively mature neural network model, which has a history of many years since it was proposed. BPNN has a simple structure and fast calculation response, which is very suitable for engineering problems with few influencing factors in the robot polishing process.

The structure of BPNN is shown in Fig. 8, which includes an input layer, multiple hidden layers, and an output layer. The training process follows the error backpropagation to continuously adjust the network weights and bias using the gradient descent method, ultimately minimizing the network error.

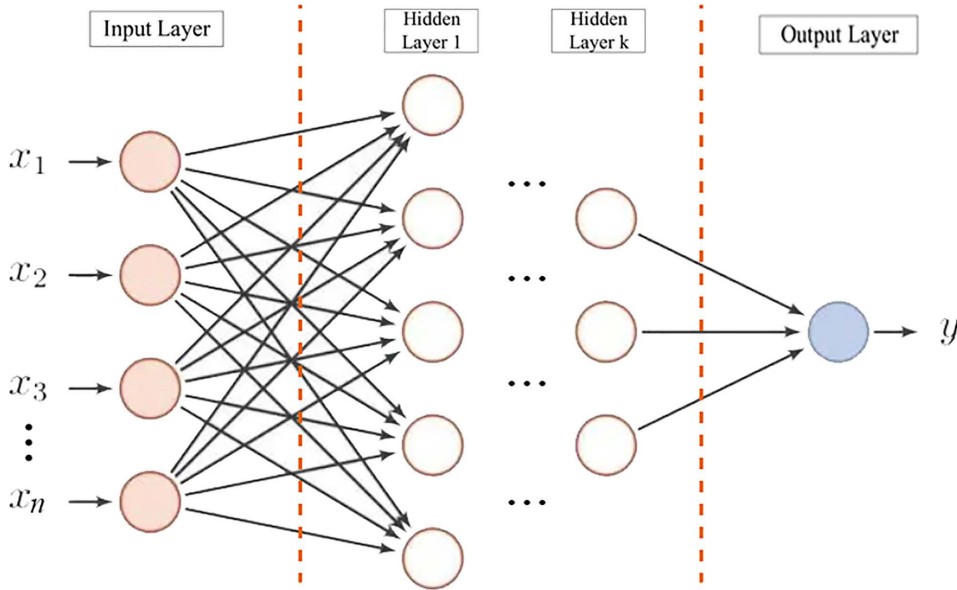

**Fig 8. Backpropagation neural network structure.**

In the BPNN model, let the weight between node $i$ and node $j$ be $j_{ij}$, the bias of node $j$ be $j_j$, and the output value of each node be $x_j$. The calculation of $x_j$ is as shown in formula (40), where $f$ is the activation function. In this study, the Sigmod function is selected, as shown in formula (41). From this, the result of the output layer $d_j$ can be calculated, and the error is recorded as $E(w,b)$, and the error function is as shown in formula (42).

$$x_j = f(\sum\nolimits_{i=0}^{n} (w_{ij}x_i + b_j)) \tag{40}$$

$$f(x) = \frac{1}{1 + e^{-x}} \tag{41}$$

$$E(w,b) = \frac{1}{2} \sum\nolimits_{j=0}^{n-1} (d_j - y_j)^2 \tag{42}$$

To correct the weight $j_{ij}$ and bias $j_j$ of the BPNN model, the error $E(w,b)$ is corrected by the gradient descent method, respectively. The correction change of the weight $j_{ij}$ is according to formula (43), where $\eta$ is the learning rate. Similarly, the bias $j_j$ can be corrected. After obtaining the prediction results, the Root Mean Square Error (RMSE) can be used to judge the accuracy of the model, as shown in formula (44). The smaller the RMSE value, the closer the model's prediction results are to the actual values.

$$\Delta w_{ij} = -\eta \frac{\partial E(w,b)}{\partial w_{ij}} \tag{43}$$

$$RMSE = \sqrt{\frac{1}{n} \sum\nolimits_{i=1}^{n} (y_i - d_i)^2} \tag{44}$$

When trying to use the original BPNN model for predictive analysis, there are instabilities. The main influencing factors include the initial weights, initial biases, number of hidden layers, and learning factors. The initial weights and biases are randomly given values. This study combines the improved DBO algorithm to seek the optimal initial values for the weights and biases of the BPNN neural network. The integrated DBO-BPNN algorithm builds a surface roughness prediction model for robot polishing workpieces.

To further improve the prediction performance, the improved DBO algorithm is introduced to optimize key factors in BPNN, including weights, biases, learning rates, and the number of network layers. The workflow diagram is shown in Fig 9.

In this study, in order to further optimize the influence of four key factors on surface roughness in the polishing process, we combined the prediction results and used the response surface optimization analysis method to optimize the values of these factors. On the basis of establishing the BPNN prediction model, the response surface analysis was introduced to explore the interaction effect between the four factors and determine the optimal parameter combination. The surface roughness predicted by the BPNN was used as the response value, and the relationship between the four factors and the response value was fitted using a quadratic polynomial. The equation for constructing the response surface model with a second-order polynomial is shown in formula (45).

$$Y = \beta_0 + \sum\nolimits_{i=1}^{n} \beta_i X_i + \sum\nolimits_{i=1}^{n} \beta_{ii} X_i^2 + \sum\nolimits_{i=1,i<j}^{n} \beta_{ij} X_i X_j + \varepsilon \tag{45}$$

Where, $Y$ is the response variable (surface roughness). $X_i$ and $X_j$ are input variables (four factors of polishing). $\beta_0$, $\beta_i$, $\beta_{ii}$, and $\beta_{ij}$ are the intercept term, the linear coefficient of variable $X_i$, the quadratic coefficient of variable $X_i$, the interaction coefficient between variables $X_i$ and $X_j$, and $\varepsilon$ is the error term. The BPNN model for roughness prediction is configured

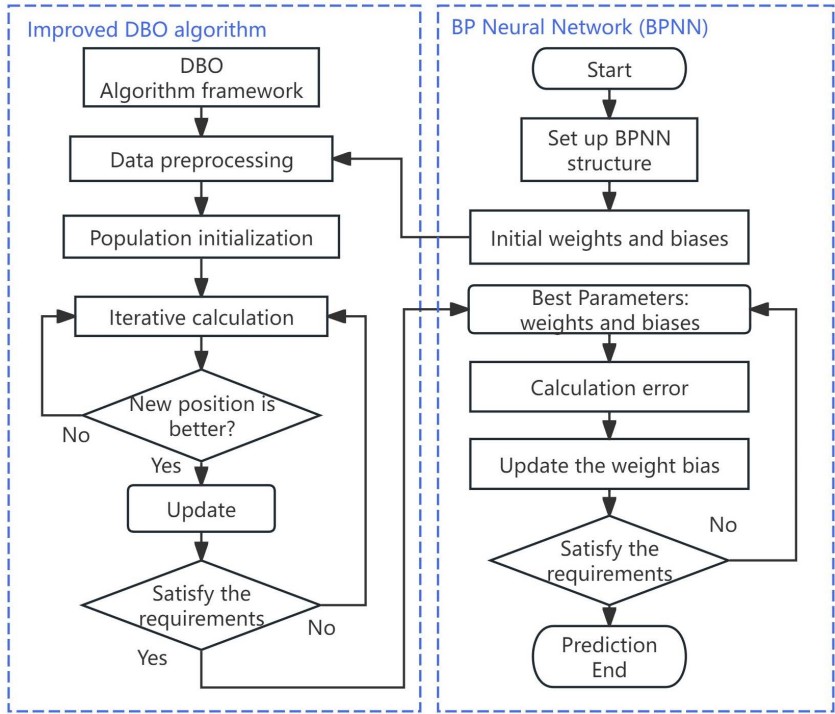

**Fig 9. Flowchart of DBO-BPNN prediction.**

with 4 input neurons, 2 hidden layers (12 and 8 neurons, respectively), and 1 output neuron. The activation function is Sigmoid, and the network is trained using the Levenberg-Marquardt algorithm with a learning rate of 0.03 and a maximum number of training times of 500. These settings are optimized using grid search and 5-fold cross-validation.

In two-dimensional or three-dimensional graphs, response surfaces can be drawn to show how changes in the two factors of robotic polishing affect the roughness, and the optimal robotic polishing process parameters can be selected by observing the synergistic relationship between the variables.

## 3. Results and discussion

This section validates the correctness of the adaptive force-position-speed collaborative robot polishing process planning proposed in this study. Research on the robot polishing process requires physical experiments and corresponding simulation experiments. Relevant robot polishing experiments were carried out on the workpiece, and the results under different process parameter conditions were compared.

### 3.1. Experimental conditions

The hardware conditions of the robot polishing experiment in this study include a 6-DOF collaborative robot, a pneumatic polishing tool, different types of sandpaper, a force sensor, and workpieces of different shapes. The material of the workpiece to be polished is Aluminum Alloy Die Castings (ADC) 12. Also known as A383, this aluminum material is the most widely used choice of metal for many die-castings produced internationally.

The overall robot polishing experimental platform is shown in Fig 10. The selected grinding tool is Opt-113, with a rotation speed of 3500~15000 rpm, which can be installed at the end of the robotic arm. The selected robot is Ufactory's xArm 6 collaborative robot with 6 degrees of freedom. The relevant coordinate system is shown in Fig 11.

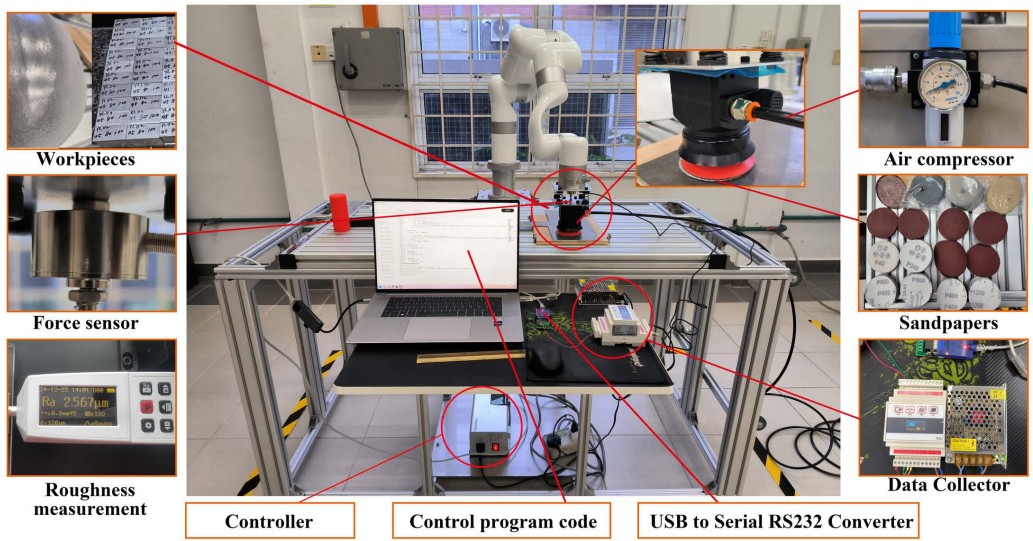

**Fig 10. Robot polishing experimental platform.**

According to the coordinate system, the improved DH parameter table of the collaborative robot polishing system can be obtained, as shown in Table 3.

### 3.2. Surface roughness modeling

The value of surface roughness $Ra$ is the surface evaluation standard to be obtained after polishing the workpiece. The relationship model between $Ra$ and process parameters is calculated concerning the material removal formula (16), which shows that the influencing factors of surface roughness $Ra$ include normal polishing pressure $Fn$, tool speed $v_t$, workpiece surface equivalent curvature radius $Re$, and feed speed $v_f$. As shown in formula (46), an equation containing a nonlinear coefficient $c_i$ can be set to represent the relationship between roughness $Ra$ and process parameters.

$$Ra = kFn^{c_1} v_t^{c_2} Re^{c_3} v_f^{c_4} \tag{46}$$

The logarithm of both sides of the nonlinear function (21) can be transformed into a linear function, and the influence coefficient can be obtained through orthogonal experiments [24]. The empirical formula of surface roughness $Ra$ when using nylon as a tool to polish an aluminum alloy curved surface workpiece is formula (47).

$$Ra = \frac{10^{0.0356} Fn^{0.6158} v_f^{0.4087}}{Re^{0.4650} v_t^{0.1035}} \tag{47}$$

Unilaterally improving surface roughness is not comprehensive, because sampling time trajectory planning must also be considered to make robot polishing more efficient. Unilaterally improving surface roughness is not comprehensive, because sampling time trajectory planning must also be considered to make robot polishing more efficient. Calculate the number of interpolation points $Num$ according to the flow chart in Fig 5. The equivalent radius of curvature at each interpolation point is different, denoted as $Rei$, $i = 1...Num$.

The equivalent radius of curvature $Re$ represents the inherent conditions of the workpiece to be polished. $Re$ and the feed speed $v_f$ determine the time $T$ for polishing the surface, which means that the surface with an equivalent radius of

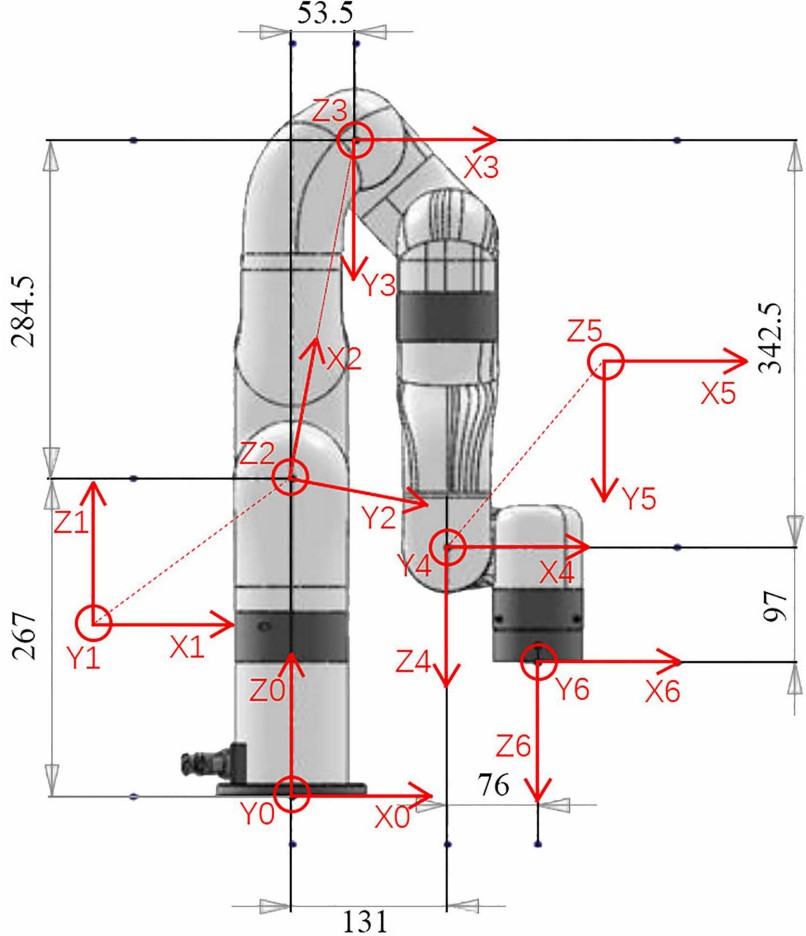

**Fig 11. Schematic diagram of the robot Modified D-H model.**

**Table 3. Robotic polishing system improves DH parameters.**

| Joint i | Length ai-1 (mm) | Twist angle αi-1 (deg) | Joint offset di (mm) | Offset (deg) | The joint angle θi limitations (deg) |
|---|---|---|---|---|---|
| 1 | 0 | 0 | 267 | 0 | [-360º, +360º] |
| 2 | 0 | −90º | 0 | −79.34995º | [-118º, 120º] |
| 3 | 289.48866 | 0 | 0 | 79.34995º | [-225º, 11º] |
| 4 | 77.5 | −90º | 342.5 | 0 | [-360º, +360º] |
| 5 | 0 | 90º | 0 | 0 | [-97º, 180º] |
| 6 | 76 | −90º | 97 | 0 | [-360º, +360º] |

$Re$ must be polished within time $T$. The robot polishing time T is used as a penalty function. The objective function that comprehensively considers the surface polishing accuracy and polishing time is the formula (48), where $u$ is the internal penalty factor of the penalty function. The polishing process must be constrained by the robot's joint motion, as shown in Equation (49).

$$min : RaT = \frac{10^{0.0356} Fn^{0.6158} v_f^{0.4087}}{Re^{0.4650} v_t^{0.1035}} + u \sum_{i=1}^{Num} \frac{Re_i^{\frac{1}{3}}}{Fn^{\frac{2}{3}} v_f} \tag{48}$$

$$S.t. \begin{cases} 10 \leq Fn \leq 40, (N) \\ 50 \leq vf \leq 300, \ (mm/min) \\ 3500 \leq vt \leq 15000, (rpm) \\ q_{imin} \leq |q_i(t)| \leq q_{imax} \\ |\dot{q}_i(t)| \leq \dot{q}_{imax} \\ |\ddot{q}_i(t)| \leq \ddot{q}_{imax} \\ q_i(t_0) = q_i(t_{end}) = 0 \\ \dot{q}_i(t_0) = \dot{q}_i(t_{end}) = 0 \\ \ddot{q}_i(t_0) = \ddot{q}_i(t_{end}) = 0 \end{cases} \tag{49}$$

The theoretical values of $F_n$, $v_t$, and $v_f$ can be obtained by optimizing the objective function (23) through a heuristic algorithm first, and then modifying it. In this study, the Dung Beetle Optimizer (DBO) [25] and Antlion Optimization Algorithm (ALO) [26], Gray Wolf Optimization Algorithm (GWO) [27], Sparrow Search Algorithm (SSA) [28], and Whale Optimization Algorithm (WOA) [29] were used for the solution. Among them, the DBO algorithm has the best performance. Fig 12 is the convergence curve of the DBO algorithm. The corresponding optimal solution is $Fn = 30 N$, $vf = 127.5895 mm/min$, $vt = 5581.058 rpm$.

Based on the theoretical optimal polishing process parameters ($Fn = 30 N$, $vf = 127.5895 mm/min$, $vt = 5581.058 rpm$) and polishing sandpaper model, the Central Composite Designs (CCD) method was selected to set up a four-factor and three-level experiment. A total of 30 sets of test results were conducted, and 5 sets of experimental data involving four factors were selected as shown in Table 4. These polishing process data are very useful for conducting surface roughness prediction research.

Experiments were conducted based on the process parameters solved by the DBO algorithm. To facilitate the verification of the results, although polishing often uses a reciprocating process of multiple cycles, the experiments are based on the results of a single polishing.

Based on multiple sets of experimental data, the BPNN is used to predict the surface roughness of the acquired data. The prediction results are shown in Fig 13, where Fig 13A is the RMSE comparison of the prediction results of the training set, and Fig 13B is the RMSE comparison of the prediction results of the test set.

The RMSE value of the training set is 0.00010898, and the RMSE value of the test set is 0.0001123. The RMSE comparison results of the training set and the test set show the accuracy of the improved DBO algorithm fused with the BPNN model. The close values between training and test sets reflect the model's generalization ability, making it reliable for surface roughness estimation in unseen conditions. The validation of the results is ensured by rigorous validation analysis, which confirms the correctness and reliability of the proposed hybrid method.

After the roughness values of different parameters are obtained by the BPNN model, the response surface and contour map are generated by interpolation fitting, as shown in Fig 14, which intuitively presents the influence of various factors and their interactions on roughness.

Fig 14A and 14B show that with the increase of $Fn$, the surface roughness shows a trend of first decreasing and then stabilizing. At the same time, the rise of tool speed $vt$ also helps to reduce the roughness, but the effect gradually weakens at higher speeds. Fig 14C and 14D show that the roughness is higher at lower $vf$, and as $vf$ increases, the roughness gradually decreases and stabilizes. The effect of the change of sandpaper type $Tp$ on the roughness shows a nonlinear trend, and large-sized sandpaper $Tp$ is more likely to obtain a better roughness value. Analysis of response surfaces reveals a synergistic interaction between tool speed and polishing force, where moderate to high values of both tend to yield optimal roughness. Feed speed shows a diminishing return effect, while sandpaper type exhibits a

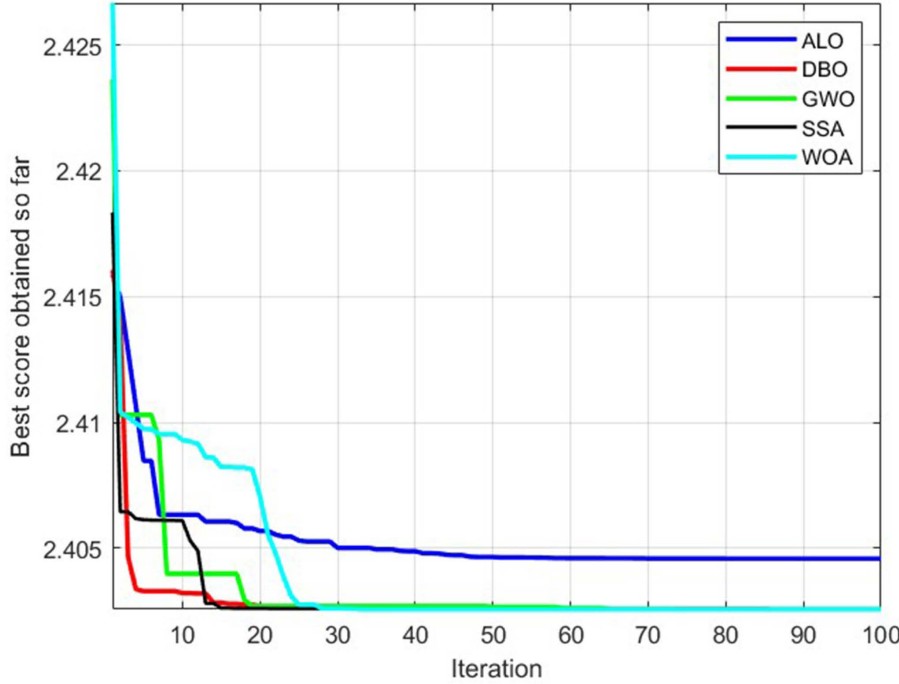

**Fig 12. Convergence curve of the DBO algorithm.**

**Table 4. Partial polishing of experimental data.**

| No. | Fn (N) | vt (r/min) | vf (mm/min) | Tp (No.) | Roughness (µm) |
|---|---|---|---|---|---|
| 1 | 15 | 5000 | 120 | 240 | 0.443 |
| 2 | 15 | 10000 | 120 | 240 | 0.416 |
| 3 | 30 | 5000 | 120 | 240 | 0.440 |
| … | … | … | … | … | … |
| 16 | 30 | 10000 | 200 | 400 | 0.383 |
| 17 | 22.5 | 5000 | 160 | 800 | 0.376 |
| … | … | … | … | … | … |
| 27 | 22.5 | 7500 | 160 | 800 | 0.339 |

nonlinear influence. Through response surface and contour analysis, the influence of each polishing parameter on the surface roughness can be intuitively observed, which provides a reliable basis for optimizing the parameter combination to improve the surface processing quality.

## 3.3. Robot polishing process parameters

The polishing process parameters selected based on the response surface optimization method were further applied to the actual operation of the robot. The finite element analysis of the robot's working condition in the polishing state was carried out using Computer-aided engineering software, providing a theoretical basis for the safety and stability of the robot's

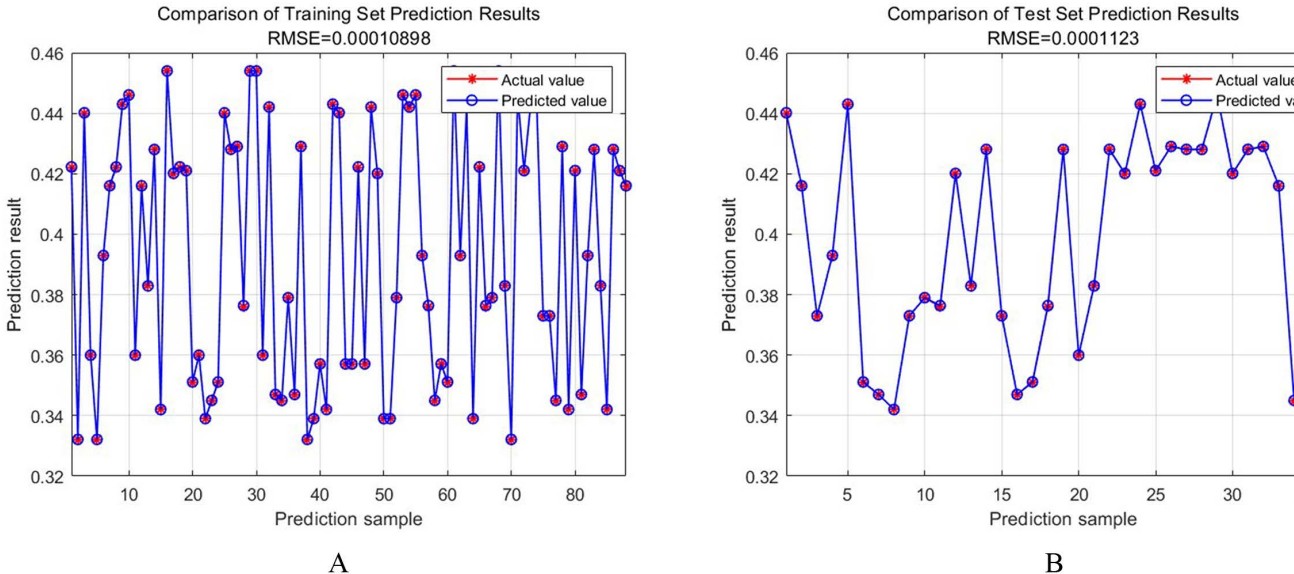

**Fig 13. Comparison of prediction results. A.** Comparison of training set prediction results. **B.** Comparison of test set prediction results.

operation. Fig 15 shows the total deformation and equivalent elastic strain of the robot under stress during the polishing process.

A finite element analysis of the robot structure was performed concerning the value of the theoretical polishing force [27]. Fig 15A and 15B respectively show the total deformation of the robot (the maximum value is 0.0403 mm) and the equivalent elastic strain (The maximum value is $1.0117 \times 10^{-5}$, which is unitless), which proves that the robot is safe and stable in this working state. This deformation is caused by the polishing reaction force on the robot, which also needs to be considered in adaptive impedance control. These findings validate the assumptions used in impedance controller design and justify the applied polishing forces.

Information about the surface to be polished can be extracted through 3D scanning or digital models. This study uses digital model surface data, which is generated using non-uniform rational B-spline surfaces. Information such as normal vectors and curvature of sampling points on the surface are easy to solve. Fig 16A is a schematic diagram of the normal vector of the surface sampling point. The normal vector of the sampling point on the surface can efficiently obtain the coordinate direction and quaternion corresponding to the point, which corresponds to (*roll, pitch, yaw*) in the base coordinate system in robot control. Fig 16B shows the polishing trajectory of the original surface.

The surface is interpolated according to the curvature adaptive algorithm proposed in this study. The blue surface in Fig 17 is the original surface, and the red surface is the surface after interpolation by the proposed method. Fig 17A shows any viewing angle, and Fig 17B is the X-Z direction viewing angle. It can be seen that the proposed algorithm is more uniform for surface interpolation, especially suitable for locations with different curvatures.

In the study, the robot was controlled to polish the curve corresponding to *X = 100* in Fig 16. The waypoints of this curve are expressed in the base coordinate system. By converting the normal vector of the waypoint into a quaternion, the coordinate direction of the robot's end effector can be solved. The Euler angle of the polishing tool is shown in Fig 18.

The deformation results of the robot under the action of polishing force reveal the structural response of the robot under polishing conditions. Subsequently, a complex surface was selected as the research object, and the position information and positive polishing force direction data of the polishing trajectory were calculated using the adaptive curvature interpolation method. This method effectively considers the geometric characteristics of the surface, generates an accurate

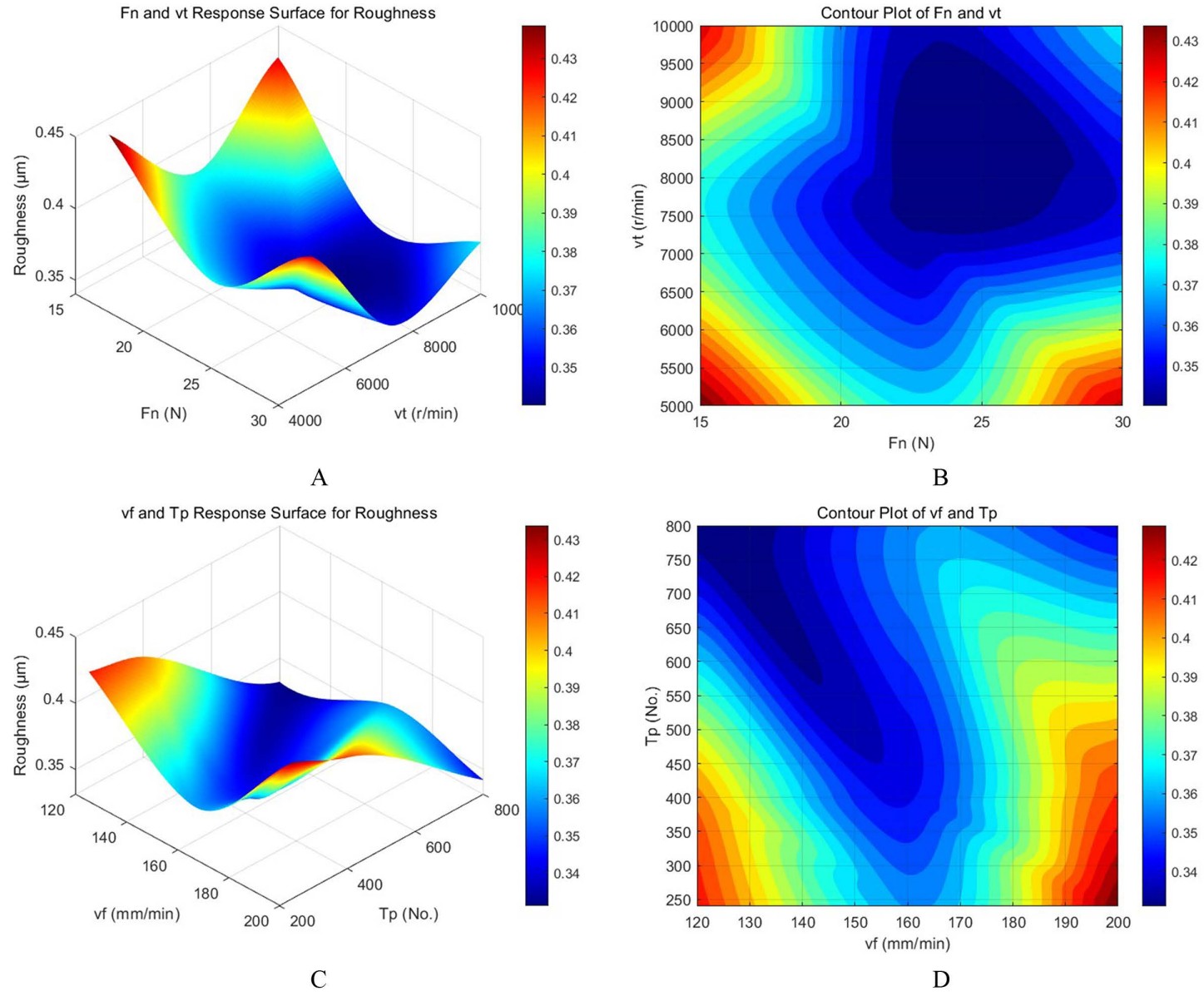

**Fig 14. Response surface and contour plots of partial polishing factors. A.** *Fn* and *vt* Response Surface for Roughness, **B.** Contour Plot of *Fn* and *vt*, C. *vf* and *Tp* Response Surface for Roughness, **D.** Contour Plot of *vf* and *Tp*.

polishing path and positive pressure distribution, and provides optimized process parameters for the robot to perform efficient and stable polishing operations.

## 3.4. Robot polishing experiments

This section aims to experimentally validate the robot polishing operation based on the polishing process parameters optimized in the previous. The polishing experiment is based on the determined four polishing elements (polishing force, tool speed, feed speed, and sandpaper type), and combines the trajectory point position and positive polishing force direction

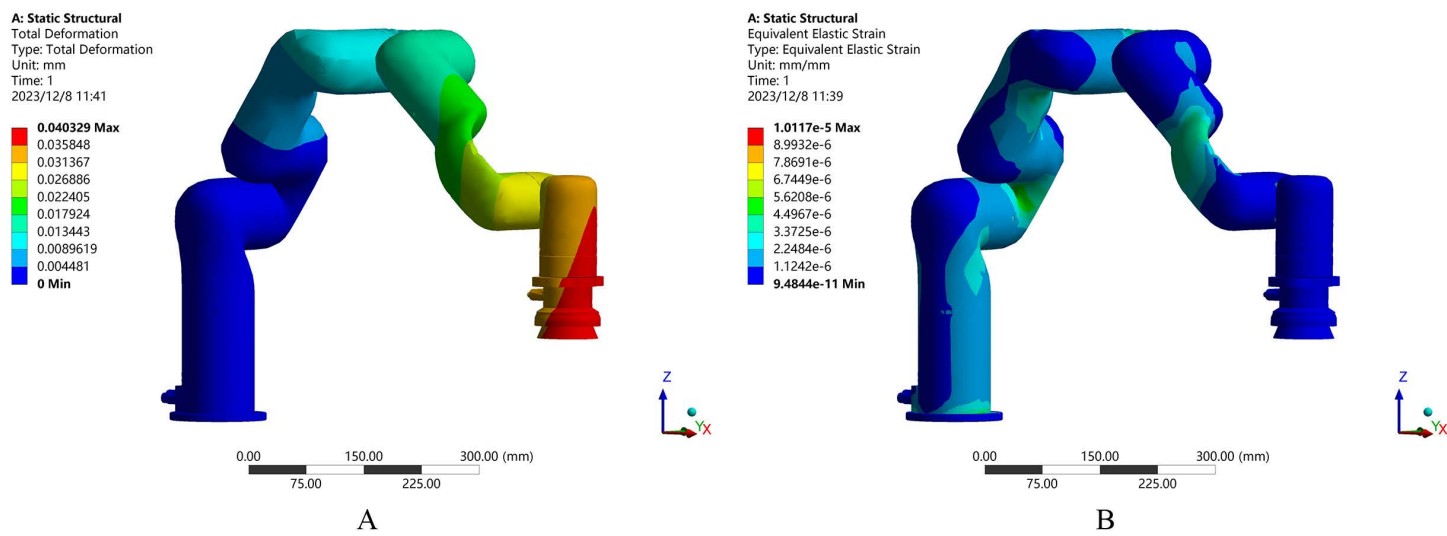

**Fig 15. Finite element analysis of the polishing process.** A. Total deformation, B. Equivalent elastic strain.

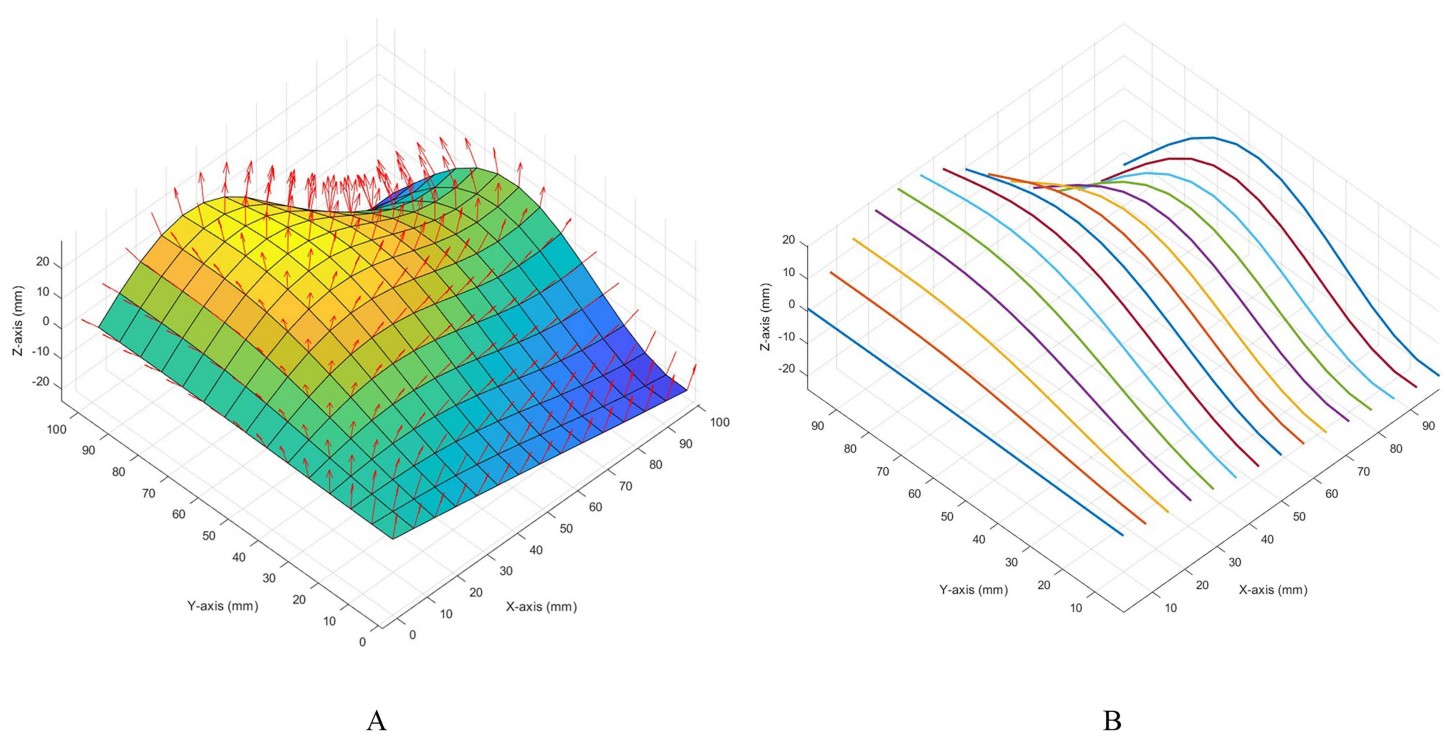

**Fig 16. The original surface for polished. A.** Surface normal vector, **B.** Polishing trajectory.

values generated by the adaptive curvature interpolation method to construct a complete polishing process. By reasonably configuring the process parameters and path planning, the precise coverage and uniform force of the polishing trajectory on the complex surface are guaranteed.

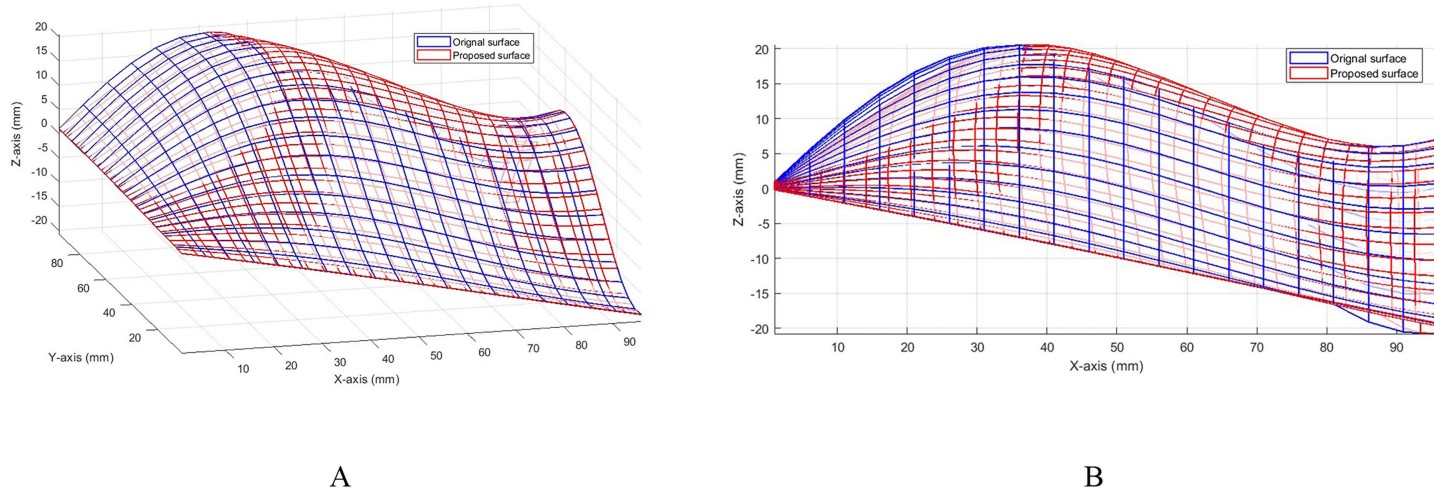

**Fig 17. Adaptive curvature interpolated surfaces, A. Comparison chart, B. Local (X-Z) comparison chart.**

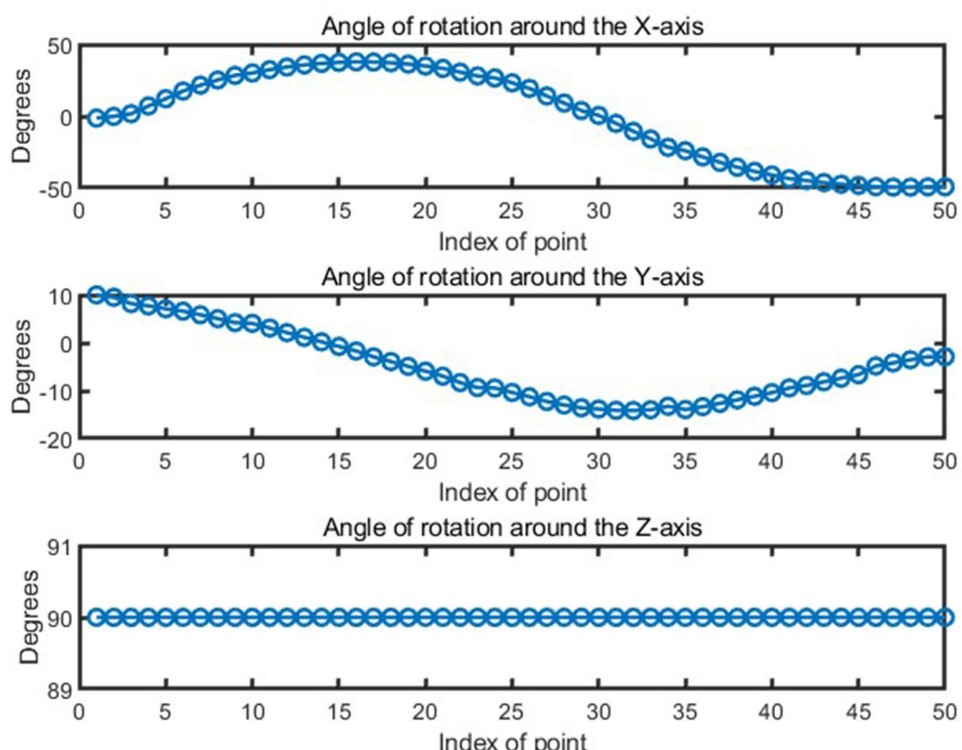

**Fig 18. Orientation of waypoints during robotic polishing.**

During the experiment, the adaptive impedance control method is introduced to adjust the robot's tracking accuracy and the stability of the applied force on the polishing trajectory in real time. To further verify the joint parameters of the robot polishing process, the angle, angular velocity, angular acceleration, and angular jerk of the 6 joints were tracked and

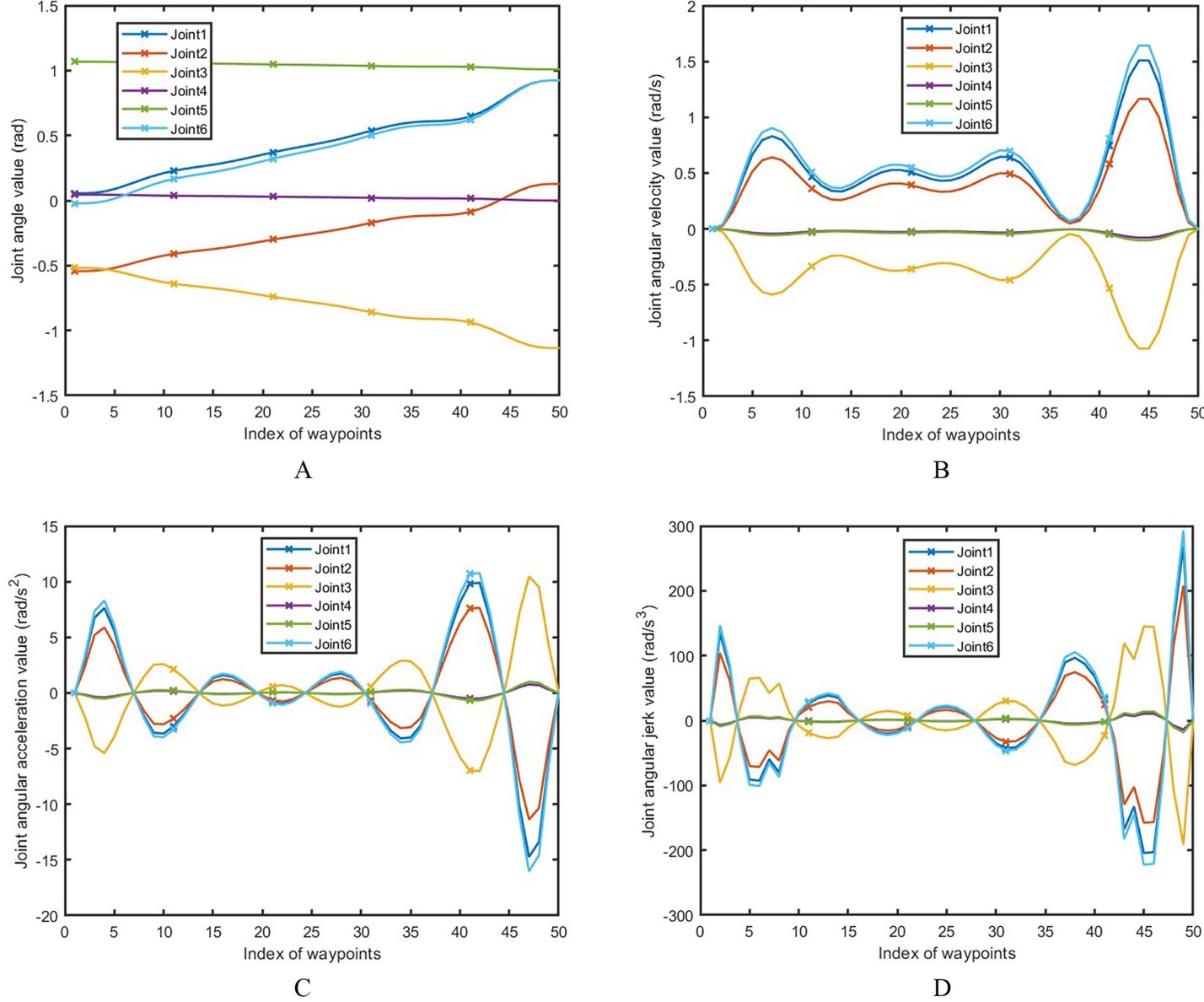

**Fig 19. Joint parameters for the polishing process. A.** Joint angle value, **B.** Joint angular velocity value, **C.** Joint angular acceleration value, **D.** Joint angular jerk value.

calculated [28], and their values correspond to Fig 19A-D respectively. It can be clarified that the robotic polishing process is continuously smooth.

The angular velocity and angular acceleration of the joint remain smooth and continuous, as evidenced by the absence of abrupt changes in angular acceleration and jerk. This approach dynamically senses variations in environmental forces during the polishing process, autonomously adjusts the flexibility and damping characteristics of the robot's joints, and ensures that the polishing force consistently fluctuates within the target range. As a result, it effectively achieves the coordinated integration of force control and position control.

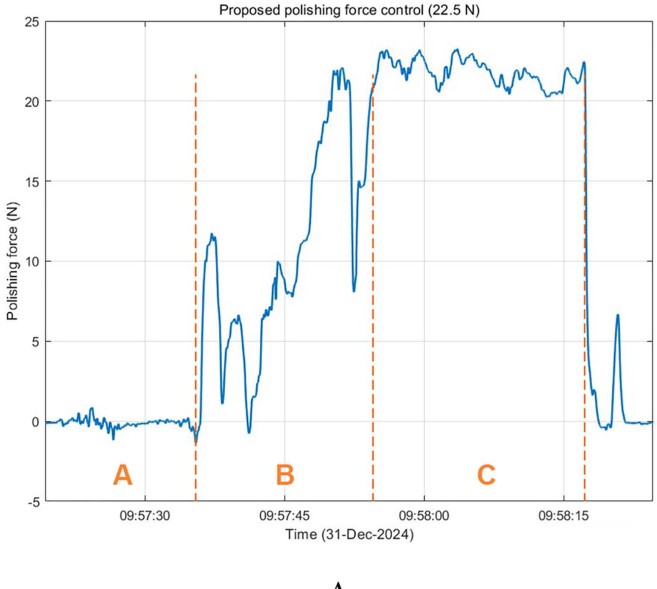

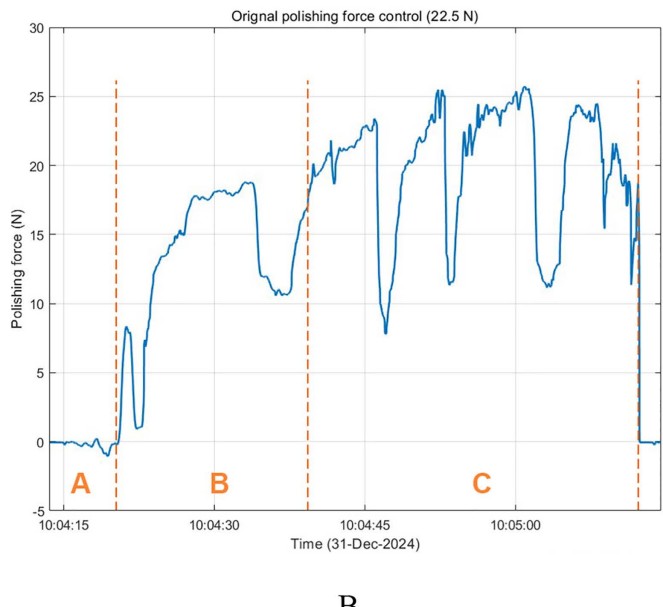

A B

**Fig 20. Measurement comparison of polishing force control.**

To validate the stability and accuracy of the proposed force control strategy during the polishing process, the performance of the proposed polishing force control was compared with that of the original polishing force control. Fig 20 shows the force feedback signal comparison between the proposed polishing force control and the original polishing force control. Zone A represents the period when the tool does not contact the workpiece before polishing, zone B represents the period when the tool contacts until the target force is reached, and zone C represents the period when the tool is stably polishing according to the preset polishing force. It can be seen that the proposed strategy significantly reduces the fluctuation of force control, making the polishing force closer to the target value (22.5 N), and the response of each stage is smoother. The original control strategy shows obvious force fluctuations, especially in the polishing process of stage C, where force mutations and overshoots are more frequent.

Compared to the original control method, the proposed impedance-based force control significantly reduces the overshoot and oscillation in steady-state polishing (zone C), allowing the actual force to closely track the desired value (22.5 N). This result validates the robustness and adaptability of the proposed controller.

When conducting the robot polishing experiment, to validate the effectiveness of the optimized polishing process parameters, the optimal parameters were selected to perform polishing operations on the specified workpiece. After the experiment was completed, the surface roughness value of the workpiece was measured to evaluate the polishing effect. The calculation of roughness $Ra$ and root mean square roughness $Rq$ is as shown in Equation (50).

$$\begin{cases} R_a = \frac{1}{n} \sum_{i=1}^{n} \left| Z(x) - F(x) \right| \\ R_q = \sqrt{\frac{\sum (Z(x) - F(x))^2}{n}} \end{cases}$$

(50)

Where, $Ra$ represents the arithmetic mean roughness, which is defined as the average value of the absolute value of the surface deviation, reflecting the overall level of the surface micro-geometric characteristics; $Rq$ is the root mean square roughness, which is obtained by taking the square root of the square average of the surface deviation, and can more sensitively reflect the changes in surface ups and downs. By measuring and calculating these two key parameters, the

Table 5. Polishing experimental data (*μm*).

| No. | Original surface | | Original polishing | | Proposed polishing | |
|---|---|---|---|---|---|---|
| | *Ra* | *Rq* | *Ra* | *Rq* | *Ra* | *Rq* |
| 1 | 4.1143 | 4.8832 | 0.36441 | 0.39376 | 0.31073 | 0.33628 |
| 2 | 6.7788 | 6.3948 | 0.65485 | 0.72599 | 0.51509 | 0.59882 |
| 3 | 1.5994 | 1.7833 | 0.72892 | 0.79831 | 0.48509 | 0.51628 |
| 4 | 3.4798 | 4.2192 | 0.36970 | 0.42460 | 0.36681 | 0.39454 |

surface quality after robot polishing can be quantitatively analyzed, providing a reliable basis for subsequent process optimization and performance evaluation.

The surface roughness *Ra* and root mean square roughness *Rq* values calculated by formula (50) at multiple positions of the experimental measurement are shown in Table 5. It can be concluded that the proposed curvature adaptive interpolation algorithm and adaptive impedance control have a positive significance for improving the surface roughness of robot polishing. The roughness (μm) has been significantly improved. Combined with the four measurement segments, the average optimization range is 20.79%.

The No. 3 parameter (vt = 6000, N = 22.5N, vf = 120, type = 320) is selected to polish a section of the trajectory to compare the proposed algorithm with the original algorithm. The comparison of results is shown in Fig 21.

The results show that the surface quality achieved by the proposed algorithm is significantly improved compared to the original polished profile and the unpolished surface. Specifically, for the surface profile of the workpiece, the roughness

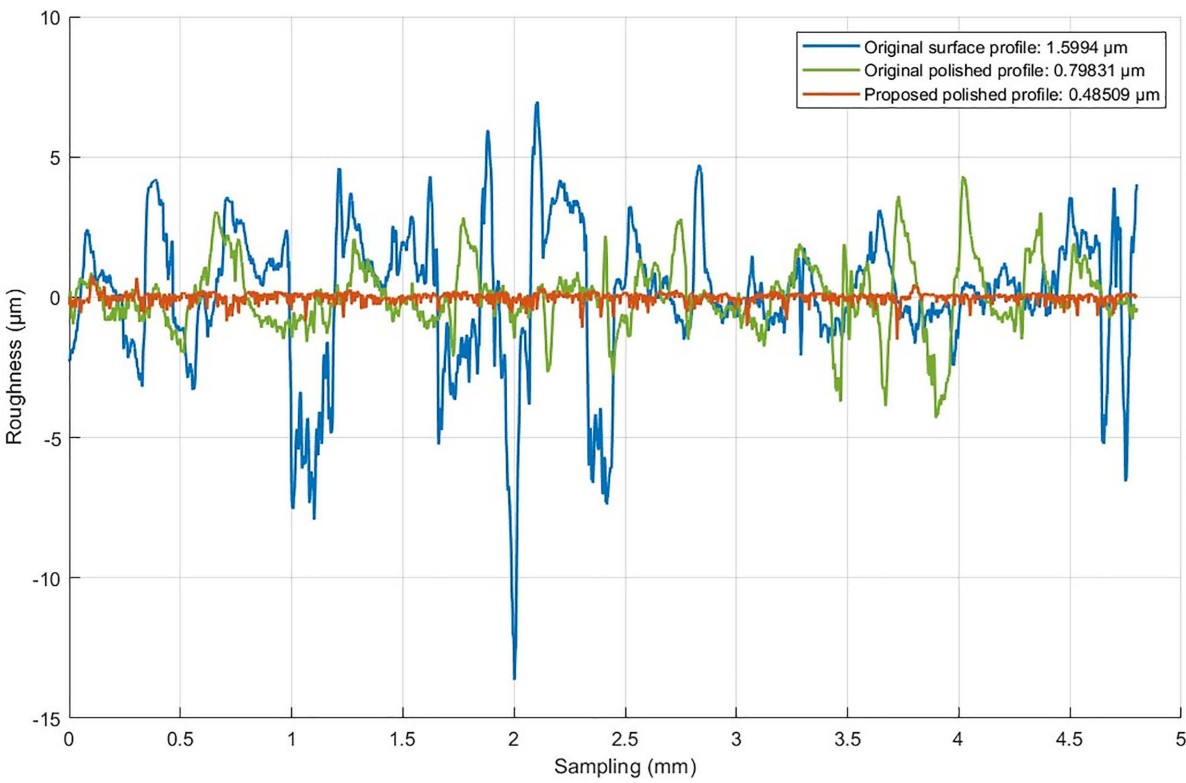

**Fig 21. Comparison of surface profiles.**

is reduced to an average of 0.52947 μm after polishing using the original algorithm, indicating an improvement in surface quality. However, the proposed algorithm further optimized the surface, reducing the roughness to an average of 0.41943 μm, which improved the surface performance by 20.79%. This demonstrates excellent consistency and smoother machining results, which is particularly important for high-precision parts.

This comparison highlights the effectiveness of the proposed algorithm in achieving smoother and more consistent surface profiles. By effectively addressing the dynamic changes in the polishing process, the proposed algorithm ensures a more uniform distribution of polishing forces and a more precise trajectory, which helps to improve surface quality. These findings highlight the potential of the proposed method for practical applications in robotic polishing tasks that require high-precision surface finishing.

## 4. Conclusion

This study developed an adaptive force-position-speed collaborative process planning framework for robot polishing, which solved the challenges of surface roughness and process stability in automated manufacturing. The main research focuses on the material removal mechanism and polishing process parameter optimization, curvature adaptive polishing trajectory, and robot adaptive impedance control.

A material removal model was constructed based on Preston's theory, and the process parameters were optimized by the improved DBO algorithm. The obtained parameters effectively guided the polishing experiment, and a four-factor three-level experiment was designed to carry out manual polishing. The four factors include polishing positive force, tool speed, feed speed, and sandpaper type. The improved DBO-BPNN model established using the data set of the polishing experiment can predict the roughness results. The RMSE of the training set and test set of the DBO-BPNN prediction model are 0.00010898 and 0.0001123, respectively. In this paper, a second-order response surface was attempted to be established to more intuitively analyze the influence of each factor on the surface roughness of the workpiece. For any two of the four factors, the influence on the roughness is plotted, and a total of 6 groups of response surface analysis results can be obtained, which can assist in adjusting the parameters of the factors to obtain a better surface. Based on the study of these manual polishing processes, the robot polishing process parameters are proposed by combining comprehensive factors such as the force deformation of the robot.

This paper proposes a novel curvature adaptive interpolation method to generate trajectories that ensure surface coverage and uniform polishing, especially for complex curvatures. The surface features to be polished are extracted through 3D scanning or digital models, and then the adaptive curvature trajectory is planned. According to the normal vector of the trajectory sampling point, the coordinate direction and quaternion corresponding to the point can be efficiently obtained, corresponding to roll, pitch, and yaw in the base coordinate system in robot control. The position and orientation of the trajectory obtained in this way can control the robot polishing process. The method is verified by computational simulation and experimental results, which prove its effectiveness in achieving smoother surfaces by optimizing trajectory planning.

An adaptive impedance control strategy is introduced in robot control to improve the force control performance during polishing. The change of impedance model parameters in the original impedance control method makes it difficult to ensure the transient performance of the robot, which may lead to a large contact force impact when the robot arm contacts the environment. This study proposes an adaptive impedance control method for the robot arm. The PD iteration method compensates for the estimated error of the environment, and the variable stiffness impedance control method reduces the excessive impact when the robot arm contacts the environment. The contact force error is used as the optimization target, and the impedance change is solved by the gradient descent method. The position PI controller is designed, and the RBFNN is used to approximate the errors and uncertainties caused by factors such as polishing head vibration and external disturbances in the dynamics of the robot arm. The stability of the control system of the proposed algorithm is verified by the Lyapunov method. The strategy dynamically adjusts the stiffness and compensates for environmental

changes to ensure consistent force application and improve polishing accuracy. The method has been experimentally verified, and the stability and force tracking accuracy are significantly improved.

Compared with existing robot polishing strategies, the proposed research framework shows excellent adaptability. The curvature adaptive interpolation method ensures high-precision trajectory tracking on surfaces with complex geometries, overcoming the limitations of traditional equal-step or uniform parameter trajectory planners. The adaptive impedance control strategy can effectively alleviate transient force impacts and compensate for environmental uncertainties, thereby improving force stability, surpassing the traditional fixed gain method. In addition, the DBO-BPNN model combines global optimization with predictive learning to achieve accurate surface roughness estimation under different process settings.

The effectiveness of the proposed force-position-velocity collaborative process planning in the robotic polishing process was verified by experimental validation. By integrating advanced optimization algorithms, adaptive control techniques, and computational models, the proposed framework significantly reduced surface roughness and improved the efficiency and reliability of robotic polishing. The adaptive force control strategy mitigates polishing force fluctuations, achieves smoother transitions, and maintains the target force level more effectively than conventional methods. Curvature adaptive trajectory planning ensures precise and uniform surface coverage, accommodating complex geometries with high fidelity. The optimized process parameters reduce the roughness value by 20.79% on average, demonstrating the practical advantages of the proposed algorithm and control method.

This study not only promotes an in-depth exploration of robotic polishing applications, but also provides a case for implementing adaptive collaborative processes in industrial applications. Future work teams will continue to expand robotic polishing to accommodate a wider range of materials and geometries, and continue to explore the intelligent application of robots.

## Supporting information

**S1 File. Process planning and roughness prediction for robotic polishing.**
(PDF)

## Acknowledgments

The authors would like to thank Universiti Putra Malaysia (UPM), the Malaysian Ministry of Higher Education (MOHE), and Tianshui Normal University, China for their continuous support in the research work.

## Author contributions

**Data curation:** Niu Jing.

**Formal analysis:** Mohd Idris Shah Ismail.

**Investigation:** Hafiz Rashidi Ramli.

**Software:** M.Y.M. Zuhri.

**Supervision:** Azizan As'arry.

**Validation:** Aidin Delgoshaei.

**Writing – original draft:** Ma Haohao.

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
