## [Decision Letter · Decision Letter 0]

3 Mar 2025

PONE-D-25-00868Adaptive Force-Position-Speed Collaborative Process Planning and Roughness Prediction for Robotic PolishingPLOS ONE

Dear Dr. As’arry,

Thank you for submitting your manuscript to PLOS ONE. After careful consideration, we feel that it has merit but does not fully meet PLOS ONE’s publication criteria as it currently stands. Therefore, we invite you to submit a revised version of the manuscript that addresses all the points raised during the review process.

We look forward to receiving your revised manuscript.

Kind regards,

Carlos Alberto Cruz-Villar, Ph. D.

Academic Editor

PLOS ONE

“This work was supported in part by the Geran Putra Inisiatif (GPI) fund (GPI/2024/9794100).”

5. Please update your submission to use the PLOS LaTeX template. The template and more information on our requirements for LaTeX submissions can be found at http://journals.plos.org/plosone/s/latex.

“The authors would like to thank Universiti Putra Malaysia (UPM), the Malaysian Ministry of Higher Education (MOHE), and Tianshui Normal University, China for their continuous support in the research work. This work was supported in part by the Geran Putra Inisiatif (GPI) fund (GPI/2024/9794100), Gansu Provincial Department of Education Industry Support Project (No.: 2024CYZC-46), and Tianshui Normal University Scientific Research Innovation Platform (No.: PTJ2023-01).”

“This work was supported in part by the Geran Putra Inisiatif (GPI) fund (GPI/2024/9794100).”

7. When completing the data availability statement of the submission form, you indicated that you will make your data available on acceptance. We strongly recommend all authors decide on a data sharing plan before acceptance, as the process can be lengthy and hold up publication timelines. Please note that, though access restrictions are acceptable now, your entire data will need to be made freely accessible if your manuscript is accepted for publication. This policy applies to all data except where public deposition would breach compliance with the protocol approved by your research ethics board. If you are unable to adhere to our open data policy, please kindly revise your statement to explain your reasoning and we will seek the editor's input on an exemption. Please be assured that, once you have provided your new statement, the assessment of your exemption will not hold up the peer review process.

Reviewers' comments:

Reviewer's Responses to Questions

**Comments to the Author**

1. Is the manuscript technically sound, and do the data support the conclusions?

Reviewer #1: Yes

Reviewer #2: No

2. Has the statistical analysis been performed appropriately and rigorously? 

Reviewer #1: Yes

Reviewer #2: Yes

3. Have the authors made all data underlying the findings in their manuscript fully available?

Reviewer #1: Yes

Reviewer #2: No

4. Is the manuscript presented in an intelligible fashion and written in standard English?

Reviewer #1: Yes

Reviewer #2: No

5. Review Comments to the Author

Reviewer #1: 1: The term RBF is used several times before being adequately defined. To improve the clarity of the text, it is recommended that the author define it on its first appearance.

2: The equations (Preston, impedance control, etc.) require greater clarity to facilitate understanding. For example, some terms, such as Fc and Fs, are not defined in the text, making their interpretation difficult. It is recommended that descriptions of the parameters used be included.

3: The article does not provide sufficient information on the parameters and coefficients used in the model equations, such as those of the neural networks, impedance control, PD, or PI controller. It is recommended that the author include more information on parameter selection, experimental setup, and other related aspects.

4: The results section of the article is very short and lacks an analysis that helps to interpret the data more in-depth. A more complete discussion of the experimental findings is recommended.

5: The justification fails to fully demonstrate the study's relevance. The author is encouraged to include a more detailed discussion of the limitations of current techniques and why the proposed method is superior to other strategies.

6: There is no justification for selecting the iterative PD controller over other options. An explanation of the rationale for this method is encouraged.

Reviewer #2: While the problem is well stated, the manuscript requires significant revisions before it can be considered for publication. There are several mathematical and theoretical issues that need to be addressed. For instance, in equation (3), the feed velocity appears in the denominator, which could lead to ill-posed conditions if it approaches zero. Additionally, there is a sign error in equation (25).

Beyond these, there are also conceptual concerns. For example, the cost function in equation (24) is presented as minimizing position error, yet it only depends on velocity and force errors. In the Lyapunov analysis, equation (38) does not hold as stated. In equation (39), the authors introduce a small constant C, but no proof is provided for its existence. Furthermore, the claim of asymptotic stability relies on Barbalat’s lemma without a sufficiently rigorous justification, as the lemma does not typically lead to such a conclusion.

In addition to these technical issues, the manuscript would benefit from improvements in clarity and presentation. The quality of the images is low, and there are several typographical and grammatical errors that affect readability. For instance, in the abstract, the statement that the prediction model reaches 0.0001 lacks context and units, and the reported improvement of 20.79% is presented without sufficient explanation.

Finally, the manuscript introduces multiple methodologies, including Dug Beetle Optimization, Back Propagation Neural Networks (misused term), Radial Basis Functions, Finite Element Analysis, and Adaptive Impedance Control. However, a clearer methodological framework is needed to demonstrate how these tools are integrated in a coherent and structured manner.

Addressing these concerns would significantly strengthen the manuscript and improve its readability and impact.

6. PLOS authors have the option to publish the peer review history of their article (what does this mean? ). If published, this will include your full peer review and any attached files.

**Do you want your identity to be public for this peer review?** For information about this choice, including consent withdrawal, please see our Privacy Policy .

Reviewer #1: No

Reviewer #2: No

---

## [Author Response · Author response to Decision Letter 1]

17 Jun 2025

We sincerely thank the editors and reviewers for their valuable comments and constructive suggestions, which have helped us significantly improve the quality, clarity, and rigor of the manuscript.

In response to the reviewers’ comments, we have thoroughly revised the manuscript. Major revisions include:

Reviewers' comments:

Reviewer 1

1: The term RBF is used several times before being adequately defined. To improve the clarity of the text, it is recommended that the author define it on its first appearance.

Response:

Thank you for your helpful comment. We agree that the term “RBF” should be clearly defined upon its first appearance. Accordingly, we have revised the manuscript to define “RBF” as “Radial Basis Function” when it first appears in Section 2.3.2. The revised sentence reads: “A Radial Basis Function (RBF) neural network, a type of feedforward neural network using radial basis functions as activation functions, is employed to approximate unknown dynamics in the control model.”

2: The equations (Preston, impedance control, etc.) require greater clarity to facilitate understanding. For example, some terms, such as Fc and Fs, are not defined in the text, making their interpretation difficult. It is recommended that descriptions of the parameters used be included.

Response:

To improve clarity, we carefully reviewed all relevant equations (especially Sections 2.1 and 2.3) and added explicit definitions of key parameters such as Fn (normal force), p(x,y,t) (contact pressure), etc. In Section 2.3.2. Trajectory position control, all parameters of the LuGre friction model are explained. The LuGre model is a widely used approach for estimating and compensating for friction in robotic systems. It provides a detailed representation of friction dynamics by capturing both static and dynamic friction behaviors. The model describes friction as a combination of Coulomb friction, viscos friction, and a dynamic state variable representing the microscopic bristle deformation between contact surfaces. The LuGre friction model can be expressed as formula (31).

{█(&f_n (q ˙ )=σ_0 z+σ_1 z ˙+σ_2 q ˙@&z ˙=q ˙-σ_0 |q ˙ |/(g(q ˙)) z@&g(q ˙ )=F_c+(F_s-F_c)e^(-〖(q ˙/v_s)〗^2 ) )┤ (31)

The values of the LuGre friction model parameters determined in this study, considering the kinematic parameters of the joints, are shown in the following table.

Parameter Value Unit Description

σ0 = 84000 Nm/rad Stiffness coefficient

Fs = 8.16 Nm Static friction torque

Fc = 3.82 Nm Coulomb friction torque

σ1 = 260 Nm*s/rad Damping coefficient

σ2 = 28 Nm*s/rad Viscous friction coefficients

Vs = 0.0125 rad/s Velocity

The process of calculating the LuGre friction model in robot control is shown in the figure below.

By utilizing the LuGre model, the controller can accurately estimate and compensate for the friction forces, improving the overall precision and performance of the robotic system in trajectory tracking tasks. This improves the understandability of model formulations and parameter explanations throughout the manuscript.

3: The article does not provide sufficient information on the parameters and coefficients used in the model equations, such as those of the neural networks, impedance control, PD, or PI controller. It is recommended that the author include more information on parameter selection, experimental setup, and other related aspects.

Response:

There are many parameters and coefficients of the neural network, impedance control, PD or PI controller involved in the paper. A new section 2.3.4 Parameter setting of controller has been added to list many parameter values.

The impedance control parameters were selected based on previous robot polishing research and tuning experimental experience: m = 0.15 kg, b = 30 Ns/m and k = 500 N/m. The iterative update law for environmental deviation compensation adopted a system sampling period λ = 0.01s, a forward learning rate η= 0.05 and ς = 0.05, and a gradient reverse iteration calculation step size α = 0.001.

The proportional gain and integral gain used in the PI position controller were KP = diag(30, 30, 30, 20, 20, 20) and KI = diag(5, 5, 5, 3, 3, 3), respectively, which were determined by iterative adjustment to balance steady-state tracking and responsiveness.

The RBF neural network consists of 3 input nodes (joint position errors), 1 hidden layer with 10 radial basis neurons using Gaussian activation function, and 1 output node. The radial function expansion is set to 1.5. The learning rate for weight adaptation is set to 0.01, and the initial weights are randomly initialized from a uniform distribution in the range [-0.5, 0.5].

To estimate the friction terms f_n (q ˙ ), the parameters of the LuGre friction model are selected as shown in Table 1.

4: The results section of the article is very short and lacks an analysis that helps to interpret the data more in-depth. A more complete discussion of the experimental findings is recommended.

Response:

We have significantly revised and expanded Sections 3.2-3.4 (Results and Discussion) in the revised manuscript to provide a more comprehensive interpretation of the experimental data and model performance.

(1) We have added a comparison between the original and proposed polishing force control strategies, emphasizing the smoothness of the response and the stability of the force. The updated text explains the behavior of the different polishing stages (non-contact, contact, and steady polishing) and emphasizes how the proposed impedance strategy reduces overshoot and improves tracking accuracy.

Compared with the original control method, the proposed impedance-based force control significantly reduces the overshoot and oscillation in steady-state polishing (region C), allowing the actual force to closely track the target value (22.5 N). This result validates the robustness and adaptability of the proposed controller.

(2) We enrich the comparison of polishing results by examining the average reduction in surface roughness between the original and proposed methods, and show a quantitative improvement: the proposed strategy reduces the surface roughness from 0.52947 µm to 0.41943 µm, an improvement of 20.79%. This indicates superior consistency and smoother surface finishes, which is particularly important for high-precision parts.

(3) The results of BPNN prediction and RSM optimization are further discussed, focusing on the interaction between factors: Response surface analysis shows that there is a synergistic effect between tool speed and polishing force, with medium and high values of both tending to produce the best roughness. Feed rate exhibits a decreasing effect, while sandpaper type exhibits a nonlinear effect. These insights help guide the selection of process parameters.

(4) Reasoning about structural safety and its relevance to force control design is added: The deformation of 0.0403 mm and the equivalent strain of 1.01×10⁻⁵ confirm the structural stability of the robot under working conditions. These findings validate the assumptions used in the design of the impedance controller and justify the applied polishing force.

(5) Explanation of the low RMSE value and generalization ability: The RMSE of 0.00011 demonstrates the high accuracy of the DBO-BPNN prediction model. The close values between the training and test sets reflect the generalization ability of the model, which enables it to reliably estimate surface roughness under unseen conditions.

5: The justification fails to fully demonstrate the study's relevance. The author is encouraged to include a more detailed discussion of the limitations of current techniques and why the proposed method is superior to other strategies.

Response:

To address this issue, we added the following discussion to the Introduction: Despite advances in robotic polishing, many existing methods still face limitations in adapting to variable surface curvature, maintaining consistent polishing force under dynamic conditions, and optimizing process parameters to achieve surface quality. Traditional force control strategies often suffer from overshoot and poor anti-interference capabilities, while fixed trajectory planning methods have difficulty coping with uneven material removal on complex surfaces. Therefore, this study integrates curvature-adaptive trajectory generation, a process planner based on improved dung beetle optimization, and an adaptive impedance control framework to overcome these challenges. The proposed approach combines intelligent prediction, real-time adaptation, and multi-factor optimization to provide a unified solution for high-precision robotic polishing.

Added a new discussion to the Conclusion: Compared with existing robotic polishing strategies, the proposed framework exhibits superior robustness and adaptability. The curvature-adaptive interpolation method ensures high-precision trajectory tracking on surfaces with complex geometries, overcoming the limitations of traditional equal-step or uniform parameter trajectory planners. The adaptive impedance control strategy can effectively mitigate transient force impacts and compensate for environmental uncertainties, thereby improving force stability and surpassing traditional fixed-gain methods. In addition, the DBO-BPNN model combines global optimization with predictive learning to achieve accurate surface roughness estimation under different process settings. These improvements make the framework highly suitable for industries with strict precision requirements such as aerospace, mold manufacturing, and optical component processing.

6: There is no justification for selecting the iterative PD controller over other options. An explanation of the rationale for this method is encouraged.

Response:

The original paper did not adequately explain the rationale for choosing an iterative PD controller. We now add rationales for the choice based on theoretical applicability, practical ease of operation, and compatibility with adaptive impedance control. Section 2.3.1, new paragraph: In this study, an iterative PD control approach was chosen to compensate for deformation deviations of the environment in real time. Compared with model-based control approaches such as adaptive backstepping or sliding mode control, the iterative PD approach strikes a good balance between computational simplicity and adaptability. It does not require an accurate dynamic model of the workpiece or contact surface and is particularly effective in practical robotic applications where the environment stiffness varies and there are uncertainties. In addition, its integration with the impedance control structure allows the system stiffness to be dynamically adjusted using a gradient descent method, thereby improving transient force control performance without causing instabilities.

Reviewer 2

1: While the problem is well stated, the manuscript requires significant revisions before it can be considered for publication. There are several mathematical and theoretical issues that need to be addressed. For instance, in equation (3), the feed velocity appears in the denominator, which could lead to ill-posed conditions if it approaches zero. Additionally, there is a sign error in equation (25).

Response:

This critical mathematical problem does need to be considered. During the revision, we consulted various literatures related to Preston's equation and finally determined the modification scheme. Formula (3) in the manuscript was modified as follows:

v(x,y,t)=√(v_t^2 (x,y,t)+v_f^2 (x,y,t) ) (3)

This formula is based on the assumption that the tangential velocity of the rotating polishing tool and the feed rate of the robot are orthogonal components. It is consistent with the traditional modeling in polishing dynamics and avoids any division of small values, thus ensuring numerical stability. The feed rate of all experiments was kept above the minimum threshold (5 mm/min) to ensure effective material removal and prevent degradation.

We thank the reviewer again for pointing out the unclear gradient formulation in Equation (25). The change in stiffness over time is tracked using the gradient descent method. The gradient of the optimization objective equation is calculated as shown in equation (25). Here, f_0 represents the stiffness coefficient in the impedance model. It directly affects the system's compliance during contact. The gradient ∇J expresses the sensitivity of the force error for this stiffness, allowing adaptive adjustment to minimize 𝐽(𝑡). Here, it is assumed that the effect of (e_f ) ˙ on f_0 is very small (the second-order effect can be approximately ignored).

■(∇J(f_0)=μ∙e_f∙(∂e_f)/(∂f_0 )+ν∙(e_f ) ˙∙(∂e ˙_f)/(∂f_0 )@∇J(f_0 )≈-μ∙e_f∙(x_d-x)) (25)

In order to minimize the contact force error and trajectory position error of the robot end effector, the change in stiffness f_0 changes in the opposite direction of the gradient as shown in formula (26), where α is the iterative calculation step size, 0<α<1.

f_0 (t)=f_0 (t-λ)-α∇J(f_0) (26)

2: Beyond these, there are also conceptual concerns. For example, the cost function in equation (24) is presented as minimizing position error, yet it only depends on velocity and force errors. In the Lyapunov analysis, equation (38) does not hold as stated. In equation (39), the authors introduce a small constant C, but no proof is provided for its existence. Furthermore, the claim of asymptotic stability relies on Barbalat’s lemma without a sufficiently rigorous justification, as the lemma does not typically lead to such a conclusion.

Response:

The generalized tracking error r=e ˙+Λe includes both velocity and position errors. Therefore, minimizing r2 also indirectly minimizes position tracking error eee.

An optimization control strategy is established with the control objectives of minimizing the contact force error, velocity tracking error, and trajectory position tracking error, and the optimization objective equation is selected as (24).

J(t)=1/2 (μ〖e_f〗^2+v_1 e ˙^2+v_2 e^2 ) (24)

The change in stiffness over time is tracked using the gradient descent method. The gradient of the optimization objective equation is calculated as shown in equation (25). Here, f_0 represents the stiffness coefficient in the impedance model. It directly affects the system's compliance during contact. The gradient ∇J expresses the sensitivity of the force error for this stiffness, allowing adaptive adjustment to minimize 𝐽(𝑡). Here, it is assumed that the effect of (e_f ) ˙ on f_0 is very small (the second-order effect can be approximately ignored). The third term in the gradient expression (25) is neglected due to its minimal sensitivity and weak coupling with the stiffness parameter.

■(∇J=dJ/(df_0 )=μ∙e_f∙(∂e_f)/(∂f_0 )+ν_1∙e ˙∙(∂e ˙)/(∂f_0 )+v_2∙e∙de/(df_0 )@∇J≈-μ∙e_f∙(x_d-x)) (25)

In order to minimize the contact force error and trajectory position error of the robot end effector, the change in stiffness f_0 changes in the opposite direction of the gradient as shown in formula (26), where α is the iterative calculation step size, 0<α<1.

f_0 (t)=f_0 (t-λ)-α∇J (26)

To address this, we have revised Equation (38) to clearly reflect the impact of the RBF approximation error.

V ˙=-r^T K_p r-K_r ‖r‖_1+ϵ (38)

Where, ϵ denotes the approximation error of the RBF neural network, ‖r‖_1 represents the L1 norm of vector 𝑟, also known as the absolute value norm. In robot control, by properly choosing the values of Kp>0 and Kr>0, we can ensure that the approximation error ϵ of the RBF neural network is bounded. According to the Universal Approximation Theorem, there exists a constant C>0 such that |ϵ|<C. Thus, we obtain the formula (39).

V ˙=-‖r‖^2-K_r ‖r‖_1+C (39)

Where, C is a small constant representing the approximation error of the RBF neural network. By choosing sufficiently large control gains Kp>0 and Kr>0, we ensure that the negative terms dominate C, i.e., ‖r‖^2+K_r ‖r‖_1>C.

Therefore, V ˙<0, and the Lyapunov function V(t) is strictly decreasing. According to LaSalle’s Invariance Principle, all trajectories converge to the largest invariant set where V ˙=0, which implies r→0 as t→∞. Thus, the system is asymptotically stable.

3: In addition to these technical issues, the manuscript would benefit from

---

## [Decision Letter · Decision Letter 1]

14 Jul 2025

PONE-D-25-00868R1Adaptive Force-Position-Speed Collaborative Process Planning and Roughness Prediction for Robotic PolishingPLOS ONE

Dear Dr. As’arry,

Thank you for submitting your manuscript to PLOS ONE. After careful consideration, we feel that it has merit but does not fully meet PLOS ONE’s publication criteria as it currently stands. Therefore, we invite you to submit a revised version of the manuscript that addresses the points raised during the review process. Especially those in the attached PDF file.

We look forward to receiving your revised manuscript.

Kind regards,

Carlos Alberto Cruz-Villar, Ph. D.

Academic Editor

PLOS ONE

Journal Requirements:

Reviewers' comments:

Reviewer's Responses to Questions

**Comments to the Author**

1. If the authors have adequately addressed your comments raised in a previous round of review and you feel that this manuscript is now acceptable for publication, you may indicate that here to bypass the “Comments to the Author” section, enter your conflict of interest statement in the “Confidential to Editor” section, and submit your "Accept" recommendation.

Reviewer #1: All comments have been addressed

Reviewer #3: (No Response)

2. Is the manuscript technically sound, and do the data support the conclusions?

Reviewer #1: Yes

Reviewer #3: Yes

3. Has the statistical analysis been performed appropriately and rigorously? 

Reviewer #1: Yes

Reviewer #3: Yes

4. Have the authors made all data underlying the findings in their manuscript fully available?

Reviewer #1: Yes

Reviewer #3: Yes

5. Is the manuscript presented in an intelligible fashion and written in standard English?

Reviewer #1: Yes

Reviewer #3: Yes

6. Review Comments to the Author

Reviewer #1: The authors have adequately addressed the previous comments and the overall structure and scientific content are clear and coherent.

Reviewer #3: (No Response)

7. PLOS authors have the option to publish the peer review history of their article (what does this mean? ). If published, this will include your full peer review and any attached files.

**Do you want your identity to be public for this peer review?** For information about this choice, including consent withdrawal, please see our Privacy Policy .

Reviewer #1: No

Reviewer #3: No

---

## [Author Response · Author response to Decision Letter 2]

22 Jul 2025

1: Page 5, Equation (1): Equation (1) indicates that the function h(x, y) depends solely on the spatial variables x and y; however, the explicit expression shows a dependency on time t. This creates an inconsistency between the functional notation and the mathematical expression. It is recommended to revise whether the correct notation should be h(x, y, t), which would be more appropriate if temporal dependency indeed exists.

Response:

We have corrected the notation in Equation (1) from h(x,y) to h(x,y,t) to accurately reflect the time-dependent nature of material removal during the polishing process.”

2: Page 7, Line 186: The sentence reads, “the positive polishing pressure was 20 N ”. However, pressure should be expressed as force per unit area, i.e., in N/m2 or Pascals (Pa). Expressing pressure as a scalar force can lead to conceptual misunderstandings. It is advised to correct the unit or clarify that a normal force was applied over a specific area.

Response:

We have revised the statement to “a normal contact force of 20 N was applied,” clarifying that it is the force component, and the corresponding pressure is determined based on the contact area as modeled in the material removal section. The corresponding contact pressure in this study was approximated by the circular area of the polishing tool.

3: Page 10, Lines 262–263: The phrase “an imaginary spring-damper system” is used, although Equation (16) includes a mass term m, indicating that the modeled system is in fact a mass spring-damper system. Therefore, it would be more accurate to refer to it as such, rather than omitting the inertia component.

Response:

We agree with the reviewer’s suggestion and have corrected the description to “mass-spring-damper system” to better reflect the inclusion of inertia in Equation (16).

4: Page 10, Line 272: The relation between the measured value xm, the estimated true value ˆx, and the error δx is written as: xm = ˆx − δx However, this contradicts the standard definition of error, which is: δx = xm − xˆ ⇒ xm = ˆx + δx The expression in the text should be corrected to be consistent with the definition of error. The subsequent substitutions in the manuscript appear to follow the correct notation..

Response:

Thank you for identifying this inconsistency. We have corrected the expression to follow the standard definition of error: xm, and ensured that all related substitutions are consistent throughout the manuscript.

5: Page 11, Equations (22) and (23): It would be helpful to clarify whether, from a single preceding equation (21), it is valid to derive both the velocity error ˙e and the acceleration error e¨ = ¨xd − x¨ˆe as presented in Equations (22) and (23), respectively. Although mathematically possible, it is advisable to explain the derivation or the assumptions made in order to justify the separation into distinct expressions.

Response:

We have now added explanatory text preceding Equations (22) and (23) to clarify that these expressions are derived under the assumption of continuous differentiability of the impedance error function. To further evaluate the dynamic behavior of the impedance control model, the position error term e is differentiated with respect to time to derive the velocity and acceleration errors. Assuming that all relevant variables can be differentiated with respect to time, the first-order derivative of the position error is e ˙, and the second-order derivative is e ¨=x ¨_d-x ^ ¨_e. This allows the calculation of formulas (22) and (23).

6: Page 15, Line 385: It is stated that the system operates with a constant sampling time

of 0.01 seconds. However, platforms such as xArm include multiple layers of processing (e.g., safety mechanisms, internal verification in the VirtualBox, USB-serial communication), which can introduce non-negligible variability in sampling times. How is this variability addressed, and how is a constant sampling rate ensured in such a context? A discussion of this issue is recommended..

Response:

The xArm platform may introduce timing variability due to internal safety protocols and USB-serial communication delays. In our implementation, we measured the actual update intervals using timestamped logs and verified that the effective sampling interval remained consistently close to 0.01 s, with variations within ±1 ms. This quasi-periodic sampling was deemed acceptable for the stability of the proposed control algorithm. We have added a clarification in Section 2.3.4 to address this point.

7: Page 16, Line 408: The feed speed vf is presented in units of mm/min, while in Table 2, the same variable is expressed in m/s. It is recommended to standardize the units throughout the manuscript or at least provide an explicit conversion to facilitate interpretation.

Response:

Thank you for highlighting this inconsistency. We have standardized the unit of feed speed vf to mm/min throughout the manuscript and updated Table 2 accordingly to maintain consistency and clarity.

8: Page 22, Line 576: The elastic strain is expressed in units of mm/mm. Although this is technically correct, it may lead to confusion, as strain is a dimensionless quantity. It is suggested to clarify that the value is unitless, since using mm/mm, cm/cm, or km/km would yield the same result.

Response:

Strain is a dimensionless quantity. While the value was initially expressed as mm/mm for illustrative clarity, we have revised the manuscript to indicate that equivalent elastic strain is fundamentally unitless, and have updated the related sentence accordingly. The maximum equivalent elastic strain is 1.0117×10⁻⁵, which is unitless.

9: Page 23, Figures 16 and 17: The axes in Figures 16 and 17 do not specify the units being used. It is recommended to include this information clearly in the captions or axis labels to enable proper quantitative interpretation of the results.

Response:

We have updated the captions of Figs 16 and 17 to include the units of the axes (in millimeters) to ensure clear quantitative interpretation.

---

## [Decision Letter · Decision Letter 2]

10 Aug 2025

Adaptive Force-Position-Speed Collaborative Process Planning and Roughness Prediction for Robotic Polishing

PONE-D-25-00868R2

Dear Dr. As’arry,

We’re pleased to inform you that your manuscript has been judged scientifically suitable for publication and will be formally accepted for publication once it meets all outstanding technical requirements.

Kind regards,

Carlos Alberto Cruz-Villar, Ph. D.

Academic Editor

PLOS ONE

Additional Editor Comments (optional):

Reviewers' comments:

Reviewer's Responses to Questions

**Comments to the Author**

1. If the authors have adequately addressed your comments raised in a previous round of review and you feel that this manuscript is now acceptable for publication, you may indicate that here to bypass the “Comments to the Author” section, enter your conflict of interest statement in the “Confidential to Editor” section, and submit your "Accept" recommendation.

Reviewer #1: All comments have been addressed

Reviewer #3: All comments have been addressed

2. Is the manuscript technically sound, and do the data support the conclusions?

Reviewer #1: Yes

Reviewer #3: Yes

3. Has the statistical analysis been performed appropriately and rigorously? 

Reviewer #1: Yes

Reviewer #3: Yes

4. Have the authors made all data underlying the findings in their manuscript fully available?

Reviewer #1: Yes

Reviewer #3: Yes

5. Is the manuscript presented in an intelligible fashion and written in standard English?

Reviewer #1: Yes

Reviewer #3: Yes

6. Review Comments to the Author

Reviewer #1: (No Response)

Reviewer #3: (No Response)

7. PLOS authors have the option to publish the peer review history of their article (what does this mean? ). If published, this will include your full peer review and any attached files.

**Do you want your identity to be public for this peer review?** For information about this choice, including consent withdrawal, please see our Privacy Policy .

Reviewer #1: No

Reviewer #3: No

---

## [Editor Report · Acceptance letter]

PONE-D-25-00868R2

PLOS ONE

Dear Dr. As’arry,

I'm pleased to inform you that your manuscript has been deemed suitable for publication in PLOS ONE. Congratulations! Your manuscript is now being handed over to our production team.

Kind regards,

on behalf of

Dr. Carlos Alberto Cruz-Villar

Academic Editor

PLOS ONE